

# GCAM-GLORY v1.0: Representing Global Reservoir Water Storage in a Multisector Human-Earth System Model

Mengqi Zhao[1], Thomas B. Wild[2], Neal T. Graham[2], Son Kim[2], Matthew Binsted[2], AFM Kamal Chowdhury[3], Siwa Msangi[4], Pralit L. Patel[2], Chris R. Vernon[1], Hassan Niazi[2], Hong-Yi Li[5], Guta Abeshu[5]

[1]Pacific Northwest National Laboratory, Richland, 99354, United States
[2]Joint Global Change Research Institute, Pacific Northwest National Laboratory, College Park, 20740, United States
[3]Earth System Science Interdisciplinary Center, University of Maryland, College Park, 20740, United States
[4]Economic Research Service, U.S. Department of Agriculture, Washington DC, 20250, United States
[5]Department of Civil and Environmental Engineering, University of Houston, Houston, 77204, United States

*Correspondence to*: Mengqi Zhao (mengqi.zhao@pnnl.gov)

**Abstract.** Reservoirs play a significant role in modifying the spatiotemporal availability of surface water to meet multi-sector human demands, despite representing a relatively small fraction of the global water budget. Yet the integrated modeling frameworks that explore the interactions among climate, land, energy, water, and socioeconomic systems at a global scale often contain limited representations of water storage dynamics that incorporate feedbacks from other systems. In this study, we implement a representation of water storage in the Global Change Analysis Model (GCAM) to enable exploration of the future role (e.g., expansion) of reservoir water storage globally in meeting demands for, and evolving in response to interactions with, the climate, land, and energy systems. GCAM represents 235 global water basins, operates at 5-year time steps, and uses supply curves to capture economic competition among renewable water (now including reservoirs), non-renewable groundwater, and desalination. Our approach consists of developing the GLObal Reservoir Yield (GLORY) model, which uses a Linear Programming (LP)-based optimization algorithm, and dynamically linking GLORY with GCAM. The new coupled GCAM-GLORY approach improves the representation of reservoir water storage in GCAM in several ways. First, the GLORY model identifies the cost to supply increasing levels of water supply from reservoir storage by considering regional physical and economic factors, such as evolving monthly reservoir inflows and demands, and the levelized cost to construct additional reservoir storage capacity. Second, by passing those costs to GCAM, GLORY enables exploring future regional reservoir expansion pathways and their response to climate and socioeconomic drivers. To guide the model toward reasonable reservoir expansion pathways, GLORY applies a diverse array of feasibility constraints related to protected land, population, water sources, and cropland. Finally, the GLORY-GCAM feedback loop allows evolving water demands from GCAM to inform GLORY, resulting in an updated supply curve at each time step, thus enabling GCAM to establish a more meaningful economic value of water. This study improves our understanding of the sensitivity of reservoir water supply to multiple physical and economic dimensions, such as sub-annual variations in climate conditions and human water demands, especially for basins experiencing socioeconomic droughts.



## 1 Introduction

Water exists in relative abundance globally, but its spatiotemporal distribution has historically posed challenges for
reliably meeting humanity's water demands. For this reason, over one-third of the world's growing population already faces
water shortage for at least one month of each year (Salehi, 2022; Oki and Kanae, 2006; Vörösmarty et al., 2000; Organization
and Fund, 2000). Humans have traditionally relied in part on surface water storage infrastructure, such as dams and reservoirs,
to manage the spatiotemporal misalignment (i.e., scarcity) of water supplies and demands, and to attenuate the effects of shocks
(e.g., droughts) (Zajac et al., 2017). Regional water scarcity can have complex societal consequences that propagate across
sectors of the economy (e.g., by constraining water available for cooling power plants and growing crops) and tele-connected
regions (e.g., through agricultural trade) (He et al., 2021). Understanding the current and potential future role of water storage
is central to our understanding of how future energy, land, and even climate systems will evolve, and in turn, how they will
impact water resources (Vanderkelen et al., 2021; Scott et al., 2016). Global multi-sector dynamic (MSD) models (Reed et al.,
2022) of the coupled human-Earth system that integrate energy-water-land-climate-socioeconomic systems are designed to
explore these interactions, yet their representations of reservoirs, and especially future reservoir storage expansion, have
remained limited (Bell et al., 2014). (See our review of reservoir representation in global MSD models in Table S1). Here we
enhance the representation of reservoir storage in a global multi-sector model, the Global Change Analysis Model (GCAM)
(Calvin et al., 2019), and demonstrate the scientific insights that can emerge as a result of this addition.

Reservoirs represent just a small stock of water within the total freshwater balance (Abbott et al., 2019), yet they play
an outsized role in satisfying human water demands (Vizina et al., 2021; Biemans et al., 2011). Over the past century, more
than 6,000 large reservoirs and dams (e.g., storage capacity > 0.1 km$^3$) have been built globally to meet growing demands for
water supply and hydropower (Lehner et al., 2011). By 2020, there were more than 58,700 registered dams worldwide, with
an aggregated storage capacity of 7,714 km$^3$ (ICOLD, 2020), which represents ≈ 20% of global annual runoff (Ghiggi et al.,
2019). If accounting for impoundments with a surface area larger than 100 m$^2$, the estimated number of dams and reservoirs
adds up to 16.7 million, with a total reservoir area of around 30,600 km$^2$ (Lehner et al., 2011). Although reservoirs only occupy
1.7% of the global inland permanent surface water extent (Liu et al., 2022), reservoirs and dams have affected more than 50%
of the world's large river systems through flow regulation, river fragmentation, and water consumption (Grill et al., 2019;
Nilsson et al., 2005). Despite the limited physical footprint occupied by reservoirs, approximately 30 to 40% of irrigated
croplands rely on reservoirs (e.g., about 265 km$^3$/year of storage-fed irrigation estimated by Schmitt et al., 2022), providing
12 to 16% of global food production (Sanmuganathan et al., 2000; World Commission on Dams, 2000). Strategic usage of
reservoir storage can improve future global sustainable irrigation and avoid the depletion of freshwater stocks and
environmental flows (Schmitt et al., 2022).

Reservoirs can be deployed to serve one or multiple purposes (e.g., flood control, irrigation, hydropower), and the
distribution of these purposes among and within the world's large river basins varies substantially. Approximately half of the
dams and reservoirs registered in the World Register of Dams (WRD) database (ICOLD, 2020) serve a single purpose, while





17.6% have multiple purposes, leaving the remainder with undefined objectives. Irrigation, hydropower, water supply, and flood control, among other purposes, represent most of the reservoirs. The Global Reservoir and Dam (GRanD) database (Lehner et al., 2011) has served as a pivotal reference, cataloguing a vast array of reservoirs and dams along with the associated primary purposes. Figure 1 categorizes reservoirs from the GranD database as hydropower and non-hydropower to demonstrate the distribution of existing storage capacity across global basins. Global hydropower and non-hydropower reservoirs have a total storage capacity of 3,745 km³ and 2,246 km³, respectively. Non-hydropower reservoirs dominate in the USA, Southern and Central Europe, Southern and Eastern Asia, North Africa, and Australia, while other regions are dominated by hydropower reservoirs. Regardless of its purpose, the most distinguishing characteristic of any large reservoir is to use storage to reshape streamflow variability to make it reliably available for human use across demand sectors and seasons (Zhou et al., 2016; Haddeland et al., 2006). Specific operational decisions on the magnitude and timing of water storage and release are dictated by the reservoir's purposes, along with other factors such as hydroclimatic conditions. Thus, depending on its purposes, a reservoir may create anywhere from multi-year (i.e., inter-annual) storage to within-year (i.e., sub-annual) redistribution of streamflow (Gaupp et al., 2015).

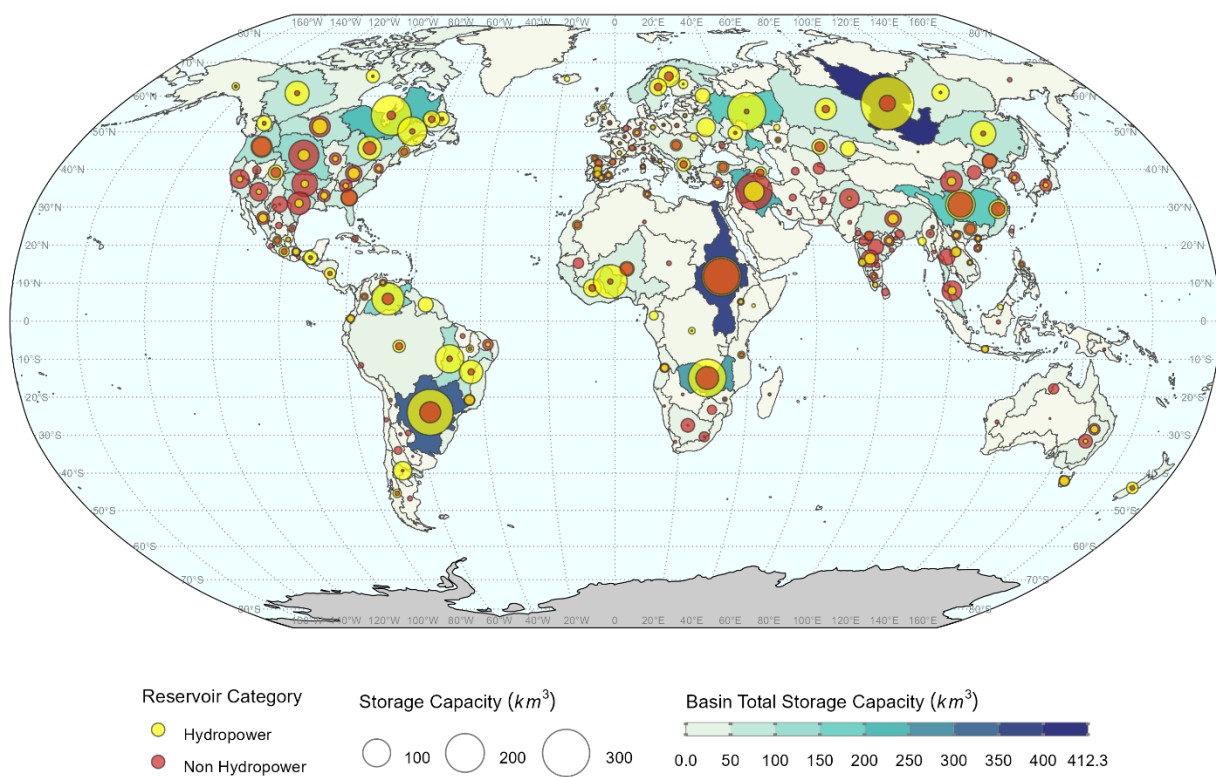

**Figure 1. Historical reservoir storage capacity (km³) for hydropower and non-hydropower reservoirs across 235 basins globally based on the GRanD v1.3 database. The yellow and red circles indicate hydropower and non-hydropower reservoir storage, respectively. The size of the circle indicates the total storage capacity for each of the two reservoir categories within the basin. The background color by which each basin is shaded quantifies the sum of the storage capacity for both reservoir categories.**



Given reservoirs supply water that drives activity in multiple sectors of the economy, incorporating reservoirs into the analysis of future multi-sector interactions among water, energy, land, and climate systems across spatiotemporal scales can bring substantial insights to our understanding of the future co-evolution of the human-Earth system (Vinca et al., 2021). Global MSD models (Reed et al., 2022) or other comparable models (see examples in Table S1), were developed for exploring inter-sectoral dynamics at regional scale with global coverage (Yoon et al., 2022; Wilson et al., 2021). Many such global MSD

models, including GCAM (Edmonds and Reilly, 1983), were initially developed to study the energy system and its emissions implications, as well as global land allocation dynamics (Keppo et al., 2021; Fisher-Vanden and Weyant, 2020). Thus, for many of these models, water has only recently emerged as central to model dynamics (e.g., for GCAM, see Kim et al., 2016). Given these models are intended for scenario-based analysis of long-term global dynamics, and are often used in stakeholder and uncertainty analysis contexts, naturally they tend to be less detailed in their representation of water resources management

(Rising, 2020). Still, even within this broad characterization of 'coarse spatiotemporal resolution', these models differ widely in their representation of water resources and reservoir management. While a detailed review of differences across models in their representation of water resources broadly is not within the scope of this article, a notable gap exists in global MSD models concerning the role of reservoirs in the co-evolution of human-Earth systems – a gap we aim to contribute to filling. To tackle this, we conducted a comparison of key differences in the representation of reservoirs across those models (Table S1).

Several studies have explored the economic impacts of water resources using GCAM by conducting uncertainty and sensitivity analysis experiments using large ensembles of global hydro-economic futures (Birnbaum et al., 2022; Dolan et al., 2021). These studies identified that the physical water scarcity and its economic impacts (e.g., on agricultural prices and revenue) are very sensitive to the representation of reservoir storage in GCAM. However, prior to our study, GCAM has traditionally used an external hydrologic model (e.g., Xanthos) to pass water availability information one-way as a boundary

condition to GCAM, while overlooking the dynamic role of existing and potential exploitable reservoir storage capacity in shaping water supply and demand dynamics. This approach makes representing future reservoir storage expansion particularly difficult, because the external model is not responding to the evolving water demand driven by sectoral interactions in GCAM.

       This paper's objective is to represent reservoir water storage in GCAM. We improve upon GCAM's representation of renewable water supply by better accounting for (1) the reliable supply potential of existing storage capacity, considering

sub-annual streamflow and demand patterns; (2) the expansion potential of reservoir storage capacity; and (3) the impact of socioeconomic change (e.g., demands) and climate conditions (e.g., socioeconomic drought) on long-term reliable water supply. Through a series of scenarios focused on the implications of socioeconomic and climate change for future water demands and reservoir expansion needs, we illuminate the new line of research questions our methodological contribution enables, and highlight challenges and opportunities in representing reservoir water dynamics within multi-sector models with

global coverage.



## 2 Methodology

In this section, we will elucidate the methodology of our novel approach for representing reservoir water storage in GCAM in four distinct sections. *Section 2.1* will provide an overview of our interactive multi-model framework and examine
its degree of coordination, communication frequency, and automation. In *Section 2.2*, we will delve into the current representation of water resource supply-demand dynamics in GCAM and delineate the aspects we intend to enhance in this study. *Section 2.3* will introduce the innovative approach we've developed for the GLObal Reservoir Yield (GLORY) model and illuminate the construction of the input data. *Section 2.4* will outline four scenarios for comparing the current and our new approach, thus providing a comprehensive illustration of the enhanced representation of water storage in GCAM from this
study.

### 2.1 Interactive Multi-Sector Dynamic Modeling Workflow

### 2.1.1 Overview

Our core contribution in this paper is GCAM-GLORY v1.0, an interactive, multi-sector dynamic modeling workflow (Fig. 2) that enables exploration of future reservoir storage expansion and its multi-sector implications at a global scale in the
context of coupled human-Earth system feedbacks. This workflow consists of multiple interacting models that capture different aspects of water resource availability, use, and infrastructure (i.e., reservoirs) at variable spatiotemporal and sectoral resolution. GCAM is central to our workflow, though we only modified inputs to GCAM (rather than GCAM's structure) in this paper. GCAM has been extensively used to answer questions across multiple disciplinary domains and has a long history of development (Calvin et al., 2019; Kim et al., 2016; Wise et al., 2009; Kim et al., 2006; Edmonds et al., 1997; Edmonds and
Reiley, 1985; Edmonds and Reilly, 1983). In our paper, its specific utility is to explore the implications of changing water resources availability (modulated by reservoirs) on the evolution of other systems (e.g., land and energy), and vice versa, at regional resolution with global coverage. GCAM represents 235 global water basins, operates at five-year time steps, and uses supply curves (i.e., the cost to supply increasing volumes of water) to capture economic competition among three categories of water supply: renewable surface water (e.g., via reservoirs), groundwater, and desalinated water. Our contribution here is
to develop a new model that is dynamically linked with GCAM to update GCAM's renewable water supply curves in each model time step to reflect the impacts of reservoir management on the cost and reliability of water supply. The interactive workflow amplifies the human-Earth system feedbacks in the context of water management in GCAM. It addresses existing research gaps by focusing on optimizing reservoir water management strategies in response to socioeconomic and climate change impacts at global to regional scales. There are a wide range of potential applications of the multi-model workflow,
such as investigating the sensitivity of renewable water supply to different drivers, and identifying potential global reservoir expansion pathways across various scenario combinations. Insight into the role of reservoirs in shaping the co-evolving energy-water-land-socioeconomic-climate system with global coverage has strong potential to inform more integrated, multi-sector strategic planning across a wide range of stakeholder groups.





In this paper, we explore two scientific questions that illustrate the advantage of this new approach: (1) How does the

cost to supply water from reservoir storage vary across global river basins, and to what extent is it shaped by hydrologic (e.g., reservoir inflows) versus economic (e.g., human water demand) characteristics?; and (2) What insights and dynamics emerge from the new approach, and to what extent can they be attributed to human-Earth system feedbacks?

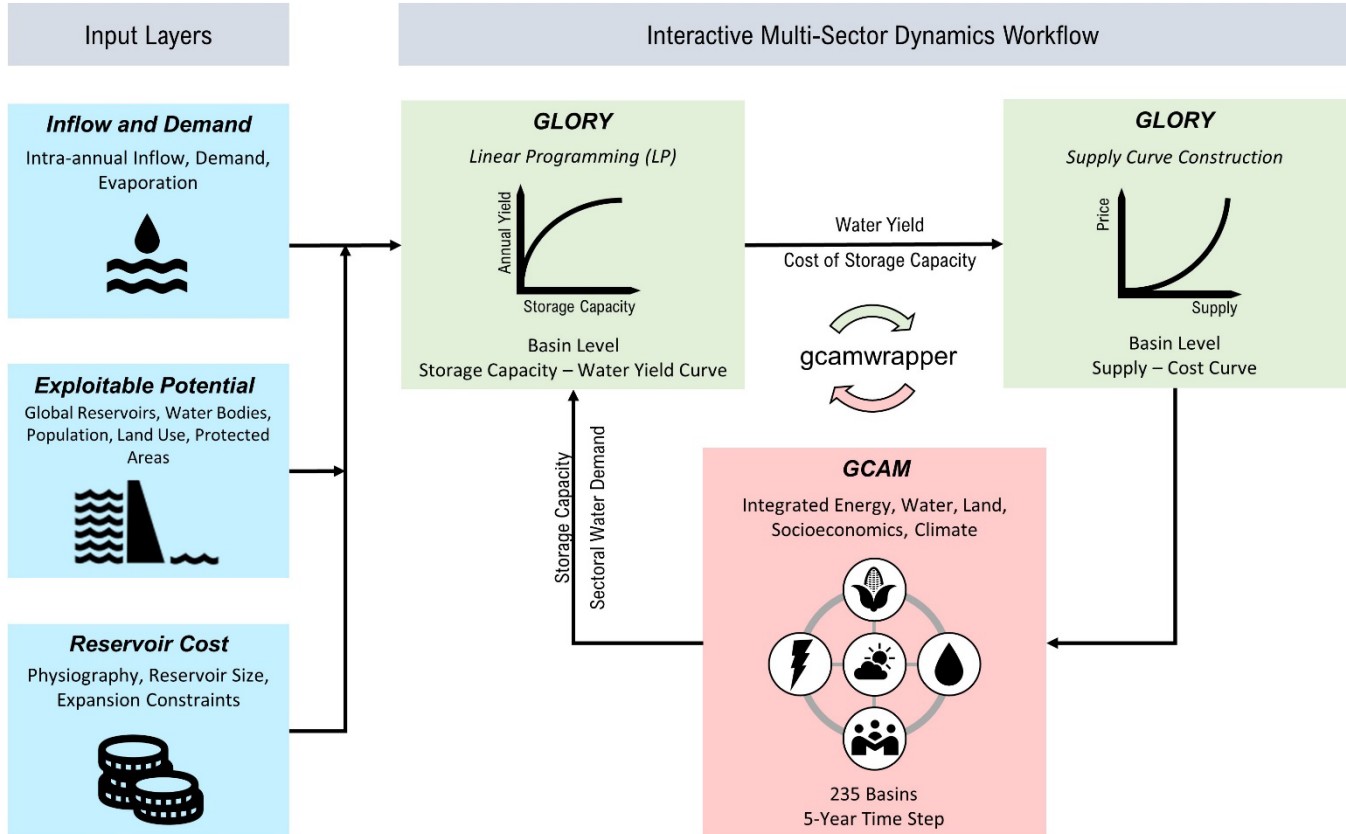

**Figure 2. Interactive modeling framework that improves the representation of reservoirs in the Global Change Analysis Model**

**(GCAM) by dynamically updating the renewable water supply curve in each time period. The GLObal Reservoir Yield (GLORY) model establishes the cost to supply water from reservoirs, receiving feedbacks from GCAM in each time step regarding sectoral water demands. GLORY ingests a wide variety of datasets, including hydrologic time series, reservoir construction costs, and land use patterns.**

The GLORY model, which we introduce for the first time in this paper, improves the current water supply curves in

GCAM by better representing the cost to deliver reliable water supply from non-hydropower reservoirs, and the physical (e.g., hydrologic inflows, reservoir exclusion zones) and economic (e.g., construction cost) dimensions that influence the cost of supply. GLORY is a Linear Programming (LP)-based model that operates at a monthly time step, but is dynamically linked with GCAM to provide GCAM, in every five-year model period, with an updated renewable water supply curve for each of 235 global water basins. The supply curve produced by GLORY specifies the unit cost to supply increasing levels of water



from reservoir storage. This supply curve establishes the basis for economic competition among alternative sources of water supply in GCAM, such as groundwater, which has its own supply curve for each basin (Turner et al., 2019). The *gcamwrapper* (Vernon et al., 2021) software coordinates interactions between GLORY and GCAM through a Python API that allows *get* and *set* operations on GCAM's internal parameters on a per period basis. After receiving the updated supply curve from GLORY (via *gcamwrapper*), GCAM executes a global simulation for the current model period, then passes water demand

outputs back to GLORY to use as input to optimization in GLORY's subsequent period. This last step advances coupled human-Earth system science by establishing interactive, fully automated feedbacks with GCAM. This paper represents the first application of *gcamwrapper* to explore human-Earth system feedbacks by coupling GCAM with an external sectoral model, though previous studies have used other tools (e.g., GCAM Fusion—a C++ based capability in GCAM that allows *gcamwrapper* to access internal parameters) for GCAM two-way coupling (Hartin et al., 2021).

**2.1.2 The Mechanics of Coupled Human-Earth System Feedbacks**

Multi-model coupling is becoming increasingly important in exploring human-Earth system interactions (Fisher-Vanden and Weyant, 2020). There are many dimensions to model coupling that can impact modeling outcomes. We characterize our workflow here with respect to its degree of coordination, communication frequency (Robinson et al., 2018), and automation.

The framework in Fig. 2 represents a high degree of (two-way) coordination, specifically between GCAM and GLORY. Coordination is the automated or manual arrangement of independently operating components and externally organized data exchange. In two-way coupling (i.e., high coordination), a software component's results render an updated state in one or more upstream components, whereas in one-way coupling (i.e., low coordination), a software component only prescribes data to one or more downstream components.

The framework in Fig. 2 represents moderate communication frequency. Communication frequency relates to how often one software component gives (or receives) data to (or from) another component. Communication frequency ranges from low/none (e.g., setting initial conditions only) to high (e.g., per time-step exchange of data). In our framework, GLORY passes a new set of supply curves to GCAM, and GCAM passes water demand and storage capacity data back to GLORY, but only for use in the subsequent time step. The models do not iterate back and forth within a time step.

The framework in Fig. 2 represents a high degree of automation. Automation is the replacement of human activity with systems or devices that enhance efficiency. Automation ranges from none (strictly manual) to a high degree of automation in which all data is exchanged virtually and none manually. The exchange of information in our workflow is fully automated.





## 2.2 The Global Change Analysis Model (GCAM) – Current Representation of Water Resources Supply-Demand Dynamics

**2.2.1 Overview**

GCAM captures the interactions among climate, land, energy, water, and socioeconomic systems at regional resolution (e.g., 235 river basins) with global coverage. GCAM is a multi-sector dynamic model and the strength of these models is the consideration of broad sectoral context and interactions, which can strongly shape the future evolution of individual systems of interest (e.g., water) (Dolan et al., 2021). This exploration of broad interactions can require sacrificing

the resolution at which individual systems (e.g., water) can be explored. This has enabled GCAM's use to study issues such as the water resource implications of climate mitigation (Hejazi et al., 2014b), the economic impacts of global water scarcity (Dolan et al., 2021), sectoral responses to water scarcity (Cui et al., 2018), future virtual water flows (Graham et al., 2023, 2020), and the regional implications of global water scarcity (Giuliani et al., 2022).

The water sector in GCAM is represented as markets with regional detail at the level of 235 large river basins (Kim

et al., 2016). As with other physical flows, such as electricity and agricultural commodities, GCAM seeks to solve for the market prices that equate water supply and demand in every water basin. The presence of 'markets' and 'prices' in GCAM is not intended to reflect literal markets on which water quantities are traded, though such markets do exist in some places. Rather, prices (and the markets that set them) allow the model to establish two key facts. First, different classes of water users (e.g., agriculture, electricity, etc.) experience different levels of access to water resources. In some places, this differentiated

access is controlled through prices—for example, the agriculture sector may receive subsidies that effectively increase its access to affordable water resources; whereas in other regions, access may be controlled by complex (and sometimes legally binding) water allocation rules. GCAM uses markets, and sectorally differentiated pricing within those markets, as a mechanism to reproduce historically observed water allocations. Note that, since the water basin is the smallest unit of analysis for GCAM, water is allocated not to individual holders of the water rights, but to a highly aggregated class of rights holders

(e.g., farmers). Second, unsatisfied demands for water have consequences—reduced production. The amount of water in use is determined by the amount physically available. Thus, when demand exceeds renewable supply in the model, runoff is reduced and the amount of water physically available in the basin is reduced to a new, lower level. The "shadow price" of water rises because end users cannot use more water than there is in the basin, leading to exploitation of more expensive water sources (e.g., deeper ground water and desalinated water), shifts to more water-efficient technologies,

reduction in crop production, or increases in trade from regions that have more affordable water supply (e.g., imports). Shadow prices are close to zero when renewable water supply exceeds demand (i.e., when cheaply available surface water does not pose a binding constraint).





### 2.2.2 Water Demand

Water demand is estimated for six sectors: irrigation, livestock, primary energy production and processing, electricity
generation, industrial, and municipal use (Hejazi et al., 2014a, b). The model includes bottom-up estimates of demands in most
sectors, based on the level of production and technology mix in each sector, which is in turn driven by socio-economic or other
factors. Future irrigation water demand depends on the evolving share of irrigated land within a particular basin-region
intersection, the individual crop classes grown on that land and their water requirements (i.e., a coefficient that establishes
water demand per unit of output), and the cost for different water sources. Water coefficients vary by crop and region. Basin-
level irrigation (i.e., blue water) demands are specified for 12 distinct crop classes (Chaturvedi et al., 2015; Mekonnen and
Hoekstra, 2011). Water demands per unit crop produced are gradually reduced over the century to reflect projected water
efficiency improvements (based on Bruinsma, 2009). Note that land use regions are subsumed into GCAM's river basins, and
thus GCAM's representation of land has similar regional characters to that of water, operating at the scale of 384 basin-regions,
which are defined as the intersections between the model's 235 water basins and 32 energy-economy regions. Besides irrigation
demands, various cooling technology options are considered with specific water demand coefficients for electricity generation,
while primary energy production estimates consider water consumption per unit energy produced for each fuel. Industrial
manufacturing's demands encompass self-supplied surface and groundwater, excluding power generation and municipal use,
which are accounted for in their respective categories. Livestock water needs rely on fixed, region-specific coefficients without
a distinction between withdrawals and consumption, while municipal use depends on population, GDP, and water prices.
Further details about water demands are documented in the GCAM documentation (Bond-Lamberty et al., 2023).

### 2.2.3 Water Supply and the Representation of Reservoir Storage

In GCAM, water supply can come from renewable water, nonrenewable (i.e., fossil) groundwater, and desalinated
water. Renewable water accessible by humans (through reservoirs and canals) is a relatively more affordable source of water
to the human system, as it does not require the substantial energy input associated with accessing groundwater and desalinated
water. Renewable water supply accounts for both direct surface water extraction and shallow groundwater pumping that draws
on recharged groundwater and captured streamflow. This study focuses on improving GCAM's renewable water supply
component only, though, as we will show, these improvements can ultimately alter future water usage behavior in other supply
categories. The process for defining a renewable water supply curve for each basin is not described in detail in previous
publications, so we describe it in detail here to better contextualize our unique contribution described in *Section 2.3*.
The current water sector in GCAM (from GCAM v5.0 to the latest v7.0 at the time of writing this paper) and GCAM's
data system, *gcamdata* (Bond-Lamberty et al., 2019), uses three key data points to establish a renewable water supply curve
(Kim et al., 2016) (Fig. 3). First, the model identifies the maximum possible quantity of water that can be exploited in a basin,
and assigns it a fixed steep price ($10/m$^3$, all dollar amounts are in 1975 USD) to indicate the unlikelihood of accessing 100%
of the basin's renewable water. In the absence of climate change impacts, the long-term mean annual flow for each basin serves





as the upper limit to available renewable supply. This value is computed using historical annual runoff, which is computed at gridded 0.5° spatial resolution in Xanthos, a Global Hydrological Model (GHM) (Liu et al., 2019; Vernon et al., 2019). This upper limit represents the maximum water supply that could be derived from a fully regulated river basin managed to maximize water yield—meaning water users can readily access the mean flow at excess cost. The high cost associated with accessing this upper limit reflects the likely high cost associated with extensive reservoir deployment, but without modeling the unique

cost of that reservoir deployment in each basin.

Only a portion of the maximum basin runoff described above is available for immediate and relatively inexpensive use, depending on environmental flow requirements and installed infrastructure for capturing, transporting, and storing water. Thus, the second data point on a given basin's cost curve establishes the quantity of water that is cheaply accessible, based on annual natural streamflow, annual baseflow, and existing reservoir storage capacity (Kim et al., 2016). GCAM assigns this

second data point a fixed low price of \$0.001/m³. This "accessible" portion of renewable water, defined in Eq. (1) below, is calculated for the majority of basins as the volume of historical annual runoff that is potentially stable (i.e., available even in dry years; Postel et al., 1996). This volume is determined by simulating the effects of both baseflows and in situ storage reservoirs included in the Global Reservoir and Dams inventory (Kim et al., 2016; Lehner et al., 2011), with an allocation of 10% of streamflow for environmental purposes.

$$QA_i^t = max\left(0, min(QT_i^t - EFR_i, QB_i^t - EFR_i + RS_i)\right) \tag{1}$$

where $QA_i^t$, $QT_i^t$ and $QB_i^t$ represent annual volumes of accessible renewable water, natural streamflow, and baseflow, respectively, in basin $i$ and year $t$; $EFR_i$ is the environmental flow requirement for each basin, and $RS_i$ is total reservoir storage capacity in each basin $i$ in the base year (2015 for the version of the model used for this publication). The $max$ in Eq. (1) makes sure the volume does not take negative values, while $min$ allows using the lesser of the net accessible stream flow ($QT_i^t - EFR_i$) and net accessible surface water in any situation, including droughts ($QB_i^t - EFR_i + RS_i$). Streamflow and

baseflow volumes are produced using Xanthos. The time series of $QA_i^t$ values from Eq. (1) is used to calculate the accessible fraction, which is defined as the ratio of the historical average annual accessible water over the historical average annual runoff. (Note that in basins for which estimates of historical groundwater depletion are available, or approximately one-fifth of basins, the accessible portion of renewable water is back-calculated from the balance of total water withdrawals and supply from groundwater depletion observed over a historical calibration period: $\frac{Withdrawals - Depletion}{Runoff}$). The water price is assumed at

\$0.001/m³ at this accessible fraction. Using these three points (i.e., \$0/m³ for no supply, \$0.001/m³ for the accessible fraction, and \$10/m³ for maximum runoff), *gcamdata* creates a 20-point curve (shown in Fig. 3), with the accessible fraction being the 10th point.



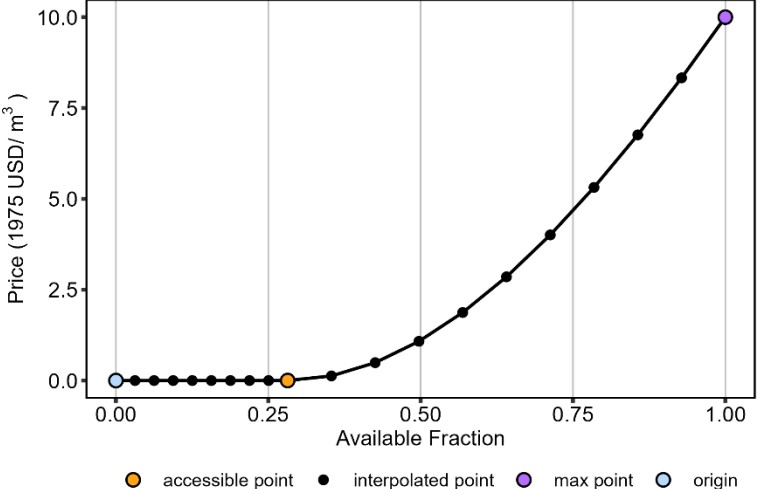

**Figure 3. An example of a renewable freshwater supply curve in GCAM, which captures the cost to supply increasing quantities of water. The figure highlights how reservoirs are currently represented in GCAM, to clarify our contribution in this paper to the representation of reservoirs.**

Reservoirs are represented in Eq. (1) (and thus in the resulting cost curve in Fig. 3), but there are four aspects of the existing approach to representing reservoirs that we seek to improve upon here. First, setting aside baseflow and environmental flow requirements momentarily, Eq. (1) essentially assumes that cumulative historical reservoir storage capacity in each basin represents the annual quantity of cheaply accessible water that reservoirs provide. However, the quantity of water that reservoirs effectively supply is not limited to the maximum physical volume of water that can be stored in reservoirs themselves (i.e., their capacity); rather, a key value of storage (in the context of water supply) is the capability to release a reliable yield for downstream use that can consistently meet sub-annually varying demand. This yield may far exceed the physical storage capacity of the reservoirs in a basin.

Second, and related to the first point, the existing approach does not dynamically simulate the capability of storage to attenuate natural inter- and intra-annual variability, including drought events. Our new approach will seek to overcome this limitation by directly simulating the sub-annual mass balance of water storage in each time period, to account for the influence of such shocks (and reservoirs' modulation of them) on reliable water supply. For example, this allows us to explore how increasingly climate-induced variability in reservoir inflows in the future may alter the quantity of water that can be reliably supplied from reservoirs.

Third, the existing approach does not account for the cost of reservoir storage itself, which is critical for establishing the cost of water supply, particularly as reservoir capacity expands in the future. Here, we capture the levelized cost to construct reservoir storage.

Finally, the existing approach does not account for the endogenous, cost-based expansion of reservoir storage over time. Previous papers have explored the implications of potential reservoir expansion pathways (Birnbaum et al., 2022; Dolan et al., 2021; Turner et al., 2019), but did so by altering the exogenously assumed accessible water fraction (Eq. 1) to reflect an





increased total storage capacity. Here reservoir expansion occurs over time based on the cost to supply water from reservoirs, and the level of expansion over time selected by GCAM is tracked by GLORY. We make these various improvements within the context of GCAM's water supply cost curve approach (i.e., without making changes to GCAM's code base) using GLORY,

which is discussed next.

## 2.3 Global Reservoir Yield (GLORY) Model

### 2.3.1 Overview

Prior to launching an optimization, GLORY processes several global input datasets (and model outputs, described in *Sections 2.3.4 to 2.3.6*) related to the hydrologic and economic characteristics of each global river basin (Fig. 2). These

characteristics include reservoir inflows, which can be produced by a GHM with global coverage over a long-term time horizon (e.g., 2020-2100) to capture climate impacts; land exclusion layers, denoting any grid cells in the world where there are significant constraints to building reservoirs (e.g., protected areas); global reservoir data, including the purpose and capacity of existing reservoirs; storage-physiography-cost relationship indicating the cost to build reservoirs at different locations; and historical monthly patterns of water demand, which helps to determine whether reservoir release patterns are producing yield

consistent with the sub-annual timing of demand. We will cover each of these input datasets in detail in the following sections. GLORY comes pre-populated with these data with global coverage, but the user can substitute their own data sets if desired (e.g., a new hydrological model's output).

Next, GLORY launches an LP-based optimization to produce a capacity-yield curve (Loucks and Van Beek, 2017) for each basin that defines the maximum reservoir discharge (i.e., yield) that increasing levels of reservoir storage capacity can

produce. (We use LP, as opposed to another technique (e.g., dynamic programming), because LP problems can be solved quickly, and many water resources problems can be formulated effectively as LP problems (Loucks and Van Beek, 2017)). Next, GLORY combines the capacity-yield curve with the levelized cost to build different levels of reservoir storage (using storage-physiography-cost relationships) to produce a renewable water supply cost curve for use by GCAM in the time period. To produce this cost curve, GLORY runs a single optimization independently for each basin. GLORY can operate using only

the input data described above; however, it can also be operated in two-way feedback mode (as we do here), wherein it receives an additional input—the sectoral water demand outputs from GCAM. These water demands, to be discussed in more detail shortly, help to quantify (1) any discrepancy that exists in the timing of monthly reservoir inflows versus demands, and (2) the corresponding storage capacity (through capacity-yield curve) used or expanded to meet the demands. Ultimately, GCAM uses the supply curve produced by GLORY as one of several inputs to make decisions on how much surface water, groundwater,

and desalinated water to deploy to meet evolving demands. GLORY's role is to identify a 'possibility curve' that defines the cost to supply surface water, whereas GCAM ultimately makes the economics-driven decisions regarding how much storage to deploy, because GCAM considers information that GLORY does not, such as the cost and availability of other sources of water (e.g., nonrenewable groundwater).



### 2.3.2 Mathematical Formulation

GLORY executes 235 unique LP-based optimizations to identify a capacity-yield curve for each of the 235 global river basins in each five-year period, building on the implementation from Liu et al., (2018). It does so by creating a single pool of water storage for each basin that aggregates each basin's distributed reservoirs into a single "virtual reservoir" that reflects the total basin storage capacity. The LP has two primary functions. Instead of seeking to reproduce historical behavior, it seeks only to sketch out a capacity-yield 'possibility curve' that reflects how much yield could be achieved if the system was

operated to maximize yield. Rather than simply maximizing annual yield, the LP forces the reservoir to adhere to monthly demand patterns. This produces a capacity-yield curve that defines the maximum quantity of water that can be annually supplied from a basin's virtual reservoir for different levels of virtual reservoir storage capacity, K, given the discrepancy between the sub-annual distribution of inflow and demand. It is this discrepancy between the scale and sub-annual distribution of reservoir inflow and demand that creates the need for storage in the first place. However, once the storage capacity reaches

a certain level, expanding storage capacity will not increase annual yield beyond the mean annual inflow over the five-year period.

Key inputs relevant to producing this capacity-yield curve are (1) the character of basin hydrology (and thus reservoir inflows and evaporation) during the five-year GCAM time period; and (2) the monthly total water demand patterns for each basin. The capacity-yield curve is dynamically updated in each GCAM period because, while storage capacity may remain the

same over time in a given basin, climate change or other influences alter hydrology and thus the renewable supply that can be achieved for a given level of capacity. GCAM dynamically supplies GLORY with the most recent (i.e., previous time period's) level of sectoral demand, which can in turn be translated into an implied GCAM monthly demand pattern for use by the LP (as discussed later). Next, the capacity-yield curve is combined with data defining (1) the cost to construct different levels of reservoir storage capacity; and (2) constraints on the maximum levels of exploitable storage capacity (e.g., accounting for

protected areas) to create an actual *cost curve* based on levelized 'overnight cost' (i.e., physical infrastructure cost). The process for constructing the cost curve will be introduced shortly, following the introduction of the generic LP formulation.

The objective of the LP is to maximize the annual water yield from a given reservoir storage capacity $K$, subject to a set of constraints. The LP consists of a monthly virtual reservoir mass balance described in Eq. (3), which allows for both environmental flows $EF_t$, return flow $RF_t$, and spillage $X_t$, in addition to the net inflow. Eq. (4) ensures a steady-state reservoir

storage, which assumes the monthly hydrologic and demand patterns repeat each year. Eq. (5) ensures reservoir storage does not fall below minimum storage or exceed maximum storage capacity. Eq. (6) to (8) denote the monthly release, inflow, and reservoir evaporation using corresponding annual volume and monthly profiles (Fig. S3, S4), described in Eq. (11) to (13). Once a virtual reservoir storage capacity, $K$, is set as input to the LP, annual reservoir evaporation $E_g$ can be determined based on the non-linear relationships between storage capacity and reservoir surface area, derived from the existing reservoirs in

each basin (Fig. S7). Eq. (6) indicates that the reservoir's monthly release must exceed monthly "demand". The sum of these monthly releases equals the annual yield that is being maximized in Eq. (2). Eq. (6) reflects that the goal is not simply to





maximize annual yield, but to do so while meeting the historical sub-annual pattern of monthly demands. Eq. (10) allows for a limit to be placed on total reservoir inflow, because of the potential for cascade reservoir systems to re-use water. The yield curve that results from this exercise is shown in Fig. S6.


$$\max_{R_t} Y_A \tag{2}$$

Subject to

$$S_{t+1} = S_t + I_t - E_t - EF_t - R_t + RF_t - X_t \tag{3}$$

$$S_{t=13} = S_{t=1} \tag{4}$$

$$S_{min} \le S_t \le K \tag{5}$$

$$R_t \ge f_t Y_A \tag{6}$$

$$I_t = p_t I_g \tag{7}$$

$$E_t = z_t E_g \tag{8}$$

$$EF_t = 0.1 I_t \tag{9}$$

$$RF_t = m(R_t + EF_t) \tag{10}$$

$$\sum_{t=1}^{12} f_t = 1 \tag{11}$$

$$\sum_{t=1}^{12} p_t = 1 \tag{12}$$

$$\sum_{t=1}^{12} z_t = 1 \tag{13}$$

where, $Y_A$ is the annual amount of water ("yield") that can be provided by a particular virtual reservoir configuration, which we seek to maximize in this LP; $S_t$ is virtual reservoir storage in month $t$ ($t = 1, \ldots, 12$); $R_t$ is the virtual reservoir release in

month $t$, and is the primary decision variable in this LP; $X_t$ is reservoir spillage, which will not be counted as part of "yield"; $E_t$ is evaporation from the virtual reservoir surface in month $t$, which is a function of total storage capacity; $I_t$ is the monthly naturalized inflow (not reflecting alteration by reservoirs or consumption) to the virtual reservoir in month $t$; $EF_t$ is the environmental flow requirement for the reservoir in month $t$, defined further in Eq. (9); $RF_t$ is the return flow in the virtual reservoir in month $t$, which represents the reusable part of the release from distributed reservoirs, defined further in Eq. (10);

$K$ is any assumed storage capacity of the virtual reservoir in the GCAM model period, between 0 and future potential storage capacity; $I_g$ and $E_g$ are the sum of mean annual basin runoff [km³/year] and the mean annual evaporation [km³/year], respectively, over the GCAM model period (5 years) of interest (e.g., 2031 – 2035 for GCAM year 2035) from all distributed reservoirs that, in total, have a summed storage capacity $K$; $f_t, p_t,$ and $z_t$ represent the fraction of annual water demand, inflow, and evaporation, respectively that occur on average in month $t$ over the GCAM model period (5 years) of interest, where the

fractions must sum to 1 over 12 months; $m$ is the fraction of flow released from distributed reservoirs ($R_t + EF_t$, but not





spillage, which is not reliably available) that's reusable in the river system, based on consumptive water use relative to demand, assumed to be 0.1 (Döll et al., 2014); $S_{min}$ is the reservoir minimum storage requirement for functional reservoir operation, assumed to be 0.

### 2.3.3 Calibration and Validation

GLORY's overarching purpose is to provide GCAM with information about the cost to supply water from reservoirs. GCAM balances water supplies and sectoral (e.g., agricultural) demands at the scale of large river basins (e.g., the entire Amazon basin). In sketching out the 'possibility space' for renewable water supply, GLORY does not seek to reproduce the exact water management and release strategies for the current and future reservoir storage installed in each basin. In fact, observed data to do so at a global scale do not exist, even for the current stock of global reservoirs (Abeshu et al., 2023).
Instead, our goal here is to identify how much water could be released from a basin's cumulative reservoir storage to meet downstream demands (should the reservoir be operated to maximize yield), given the differences in sub-annual timing of reservoir inflows and demands. This provides a reasonable upper bound that helps to constrain GCAM's water supply behavior. Rather than calibrating GLORY against observations, instead we seek to validate, and/or improve the fidelity of, the input data sets and constraints that guide the optimization procedure. For example, the hydrology model that produces reservoir inflows
is calibrated and validated. We also conduct a form of validation of GLORY's capacity-yield curves considering two aspects. The first aspect is to check that the level of water supply achieved with existing reservoir storage capacity (i.e., historical water demand data) is less than or equal to the maximum supply that GLORY suggests is possible for that same level of storage capacity. We confirm this finding in Fig. S10a with global water demand data (Huang et al., 2018), demonstrating that our approach provides a reasonable upper bound on the water supply that is possible, without attempting to represent each basin's
unique water management behavior. The second aspect is to check that the level of reservoir annual release from existing reservoir storage capacity (i.e., historical reservoir outflows) is within the range of the annual release (same as yield) that GLORY produces at the same level of storage capacity. We confirm this in Fig. S10b with reservoir outflow data ResOpsUS (Steyaert et al., 2022) for the U.S, indicating that our approach estimates reasonable release at basin scale.

### 2.3.4 Input Layer: Reservoir Inflow and Demand Data

*Natural Reservoir Water Fluxes*

     GLORY requires as input the average monthly profiles of reservoir inflow, evaporation, and demand. Monthly profiles are calculated using monthly data as the fraction of the time series variable value for each month over the sum of all 12 months. Monthly profile of inflows to, and surface potential evaporation from, the virtual reservoir is derived from the Xanthos model's monthly streamflow time series output at grid cells with existing reservoirs for the five-year GCAM period
of interest. These reservoir water flux time series can be generated for different climate change scenarios simulated using General Circulation Models (GCMs) with Representative Concentration Pathways (RCPs). The specific scenarios we explore in this paper, to be introduced shortly, include one example of a climate impact scenario. Our simulation of the mass balance





of water in the reservoir (Eq. 3), including natural shocks (e.g., droughts) in the sequence of inflows, enables us to better reflect the occurrence of drought events.

For inflow and evaporation, the monthly profiles over each 5-year window of the GCAM periods are calculated from 2020 to 2050 (using data from 2016 to 2050). Inflow and evaporation values for each month are taken as the average values over the 5-year window of the GCAM period. For example, January's runoff is the mean of five January runoff values over 2021 – 2025 for GCAM period 2025. This captures the general sub-annual pattern of inflow, along with any evolving inter-annual patterns in changing water availability. GCAM is often fed with highly smoothed data, for purposes of assessing long-

term trends in sectoral interactions, as well as to avoid solution failures due to its partial equilibrium structural design. Since GLORY is GCAM-considerate, the inputs and outputs of the GLORY model are designed to represent the averaged behavior during each GCAM period. GLORY is certainly capable of being used to explore more variable inflows, and to capture drought events, but we leave exploration of these dynamics to future studies. A basin's optimization for a particular GCAM period (e.g., 2050) is executed in the absence of any carryover of information about reservoir storage levels in the previous five-year

time period (e.g., 2045). (We feel this is a reasonable assumption, as the storage levels for many large reservoirs globally are uncorrelated across time lags exceeding five years). The purpose of the method we introduce here is to avoid reflecting unsustainable dynamics, such as progressive drawdown of reservoir levels during prolonged periods of drought. Rather, we seek to sketch out the 'possibility space' that defines how much water could be supplied if the objective was to sustainably maximize yield.


*Reservoir Water Demands*

     To execute an optimization in the first model time period (2015), GLORY requires as input the monthly fractions of historical total annual water demand that occur in each month. We use water demand data from 2005 – 2010 for this purpose (Huang et al., 2018). In future time periods (e.g., starting in 2020), the demand profile is updated using annual sectoral demand

from GCAM's previous time step from six GCAM sectors: irrigation, livestock, municipal, electricity, industry, and primary energy. To do this, sectoral monthly water demands are temporally disaggregated from GCAM's annual demands for each sector by using a historical monthly demand profile for each sector from Huang et al. (2018). This allows us to superimpose all sectoral monthly demand curves on top of one another to generate the total demand profile for the future time period. We then calculate the fraction of total demand occurring in each month, which serves as input to the LP optimization. Thus, the

raw magnitude of demands is not used by the LP, except in the sense that they provide a weighted adjustment of the monthly total water demand profile. For example, if GCAM projects that future irrigation water withdrawals will disproportionately increase over time in a particular GCAM scenario and time period, this information will be accounted for within GLORY via a monthly water demand fraction profile that shifts closer in appearance to the monthly irrigation demand profile. This may increase or decrease the yield that is possible to achieve with a given level of reservoir storage, depending on whether sub-

annual supplies and demands are misaligned in time. We address this issue in detail in the following section.





*Discrepancy Between Water Demand and Surface Water Supply*

Reservoir storage plays an important role in managing or buffering the discrepancy between natural water availability (i.e., supply, or inflows) and demands. Our approach captures the intra-annual supply-demand discrepancies caused by the hydrological and socioeconomic drivers that previously were not considered in GCAM and explores the role of reservoir storage in providing reliable water supply in different regions of the world. As captured in Eqs. (2) - (13), the capacity of a reservoir system to reliably supply water depends on the sub-annual variations and timing differences between streamflow, evaporation from the reservoir surface, and demand. Water deficit often occurs when there is not sufficient reservoir storage in place to spatiotemporally redistribute water and meet water demand. To quantify the intensity of water deficit that would exist under a shortage of reservoir storage, and therefore identify the regions that could benefit most from storage capacity expansion, we use a "socioeconomic drought" metric (Wilhite and Glantz, 1985). The socioeconomic drought intensity (SEDI) is defined as the ratio of total water deficit within a year (i.e., volume of demand that exceeds supply, when demand > supply) to the duration of the deficit, in the absence of any reservoir storage (Heidari et al., 2020).

Figure 4 shows the historical SEDI levels globally with examples of average monthly water deficit and surplus for six basins. In their unregulated states, most basins have no historical demand deficiency issues (i.e., 171 basins). However, numerous river basins around the world experience moderate-to-high deficit levels (e.g., 23 basins have *log(SEDI)* values higher than 1). Most of the basins with high SEDI values are noticeably clustered in the Middle East and Asia, though there are some regions experiencing this in the Americas. For example, in its unregulated state, the California basin has a severe water deficit from June to October because of increased demand for irrigation water, whereas abundant water resources become available during other times of the year when demands are lower. (The primary river source of water supply within the California basin includes the Sacramento River and San Joaquin River). This dynamic means that reservoir storage potentially offers significant value.



**Figure 4. Average surplus and deficit between water supply and demand during historical period in six selected basins. Surplus occurs when demand is lower than inflow (shown in green), and deficit occurs when demand is higher than inflow (shown in red). The global map shows the socioeconomic drought intensity (SEDI) as the ratio of total water deficit within a year to the duration (in months) of the deficit. The SEDI is scaled using logarithmic transformation to reduce the variance in the SEDI due to different magnitudes of water amount across basins. White color indicates there is no socioeconomic drought (supply > demands i.e., demands are always met using supply).**

### 2.3.5 Input Layer: Reservoir Storage Capacity Exploitable Potential (Land Exclusion Zones)

The potential to expand reservoir storage capacity in the future in each basin is limited to land areas where reservoirs can feasibly be constructed. Reservoir storage capacity exploitable potential, referred to henceforth as "exploitable potential", is the sum of existing reservoir storage capacity and future exploitable storage capacity at basin scale. The exploitable potential serves as a constraint for $K$ in Eq. (4). We identify the exploitable areas as the overlap among four different land exclusion





layers. Our approach is largely based on Liu et al. (2018), but we also filter out grid cells that do not contain existing water bodies suitable for the siting of reservoirs (i.e., rivers). The exclusion layers include regions of high population historically (Jones et al., 2020), historical protected areas (UNEP-WCMC and IUCN, 2022), historical cropland areas, and absence of water bodies (e.g., rivers and lakes). To ensure consistency with the available data, we assume that the exclusion layers remain unchanged from the 2010 historical benchmark. For the population exclusion layer, we use the 0.125-degree spatial resolution

population trajectory (Jones et al., 2020) from 2010 under Shared Socioeconomic Pathway (SSP) 2 "middle of the road" (O'Neill et al., 2017). To define protected areas, we use the World Database on Protected Areas (WDPA), which specifies terrestrial and marine protected areas as polygon shapes. To define cropland areas, we use 5-arcminute gridded resolution spatial land use and land cover projection from Demeter (Vernon et al., 2018; Chen et al., 2019), which disaggregates GCAM's basin-scale land allocation of 2010 to grid scale. Water bodies are identified using Global Lakes and Wetlands Database

(GLWD) Level 3 (Lehner and Döll, 2004), including land types 1, 2, and 3 (lakes, reservoirs, and rivers, respectively). All spatial data, if not already, are rasterized, aggregated or geo-referenced to the same 0.5-degree resolution. Using these four layers, we identify grid cells as exclusion cells for reservoirs (i.e., where no new reservoirs can be built) if any of the following criteria are satisfied (see Figure S5): (1) population density for the grid cell is higher than 1,244 capita per km$^2$; (2) the grid cell has protected land; (3) more than 10% of the land cover within the grid cell is crop land; and (4) no water bodies exist in

the grid cell. Note that the feasibility of constructing reservoirs can be influenced by many other factors, such as geological and seismic stability, which we do not consider here. We have chosen to emphasize these four primary constraints due to the availability of data with global coverage and our assumption of a more flexible reservoir expansion policy.

        The viable grid cells that remain (after removing cells that meet these four exclusion criteria) are grouped into three grid cell types according to relative potential of building new capacity to existing capacity and mapped in Fig. 5. Type 1

(referred to as "exploited") indicates grid cells with existing reservoirs (based on GranD database), where all feasible reservoir sites are already exploited. Type 2 (referred to as "partially exploited") indicates grid cells that contain existing reservoirs, but where more reservoirs could potentially be built. Type 3 (referred to as "unexploited") indicates grid cells with no existing reservoirs, but in which new reservoirs can potentially be built. The remaining grid cells (i.e., around 60971 transparent grid cells) have no reservoir exploitable potential. The color in each basin in Fig. 5 shows the ratio (fraction) of the excluded area

to the water surface area within the basin. The primary category that drives each basin's exclusion fraction differs by basin. For example, for regions with high fractions of excluded area that are also cold or dry (e.g., Australia or Greenland), the limited availability of surface water bodies drives the high rate of exclusion. For some basins (e.g., Zambezi basin in Africa), protected areas dominate. For other basins (e.g., Volga basin in Europe, Parnaiba basin in South America), several categories contribute equally to the total excluded area.

Several hotspots emerge (in Fig. 5) that have large numbers of grid cells where reservoirs could feasibly be built, including in South America (e.g., Orinoco, Amazon, La Plata, Sao Francisco), Africa (e.g., Congo, Nile), and North Asia (e.g., Yenisey, Lena). North America, Eastern Europe, Australia, and Eastern China are clustered with existing reservoirs and





feasible exploitable areas are much fewer. The spatial variation of exploitable potential differentiates the supply and demand potentials across the basins, which can affect long-term development of infrastructure.

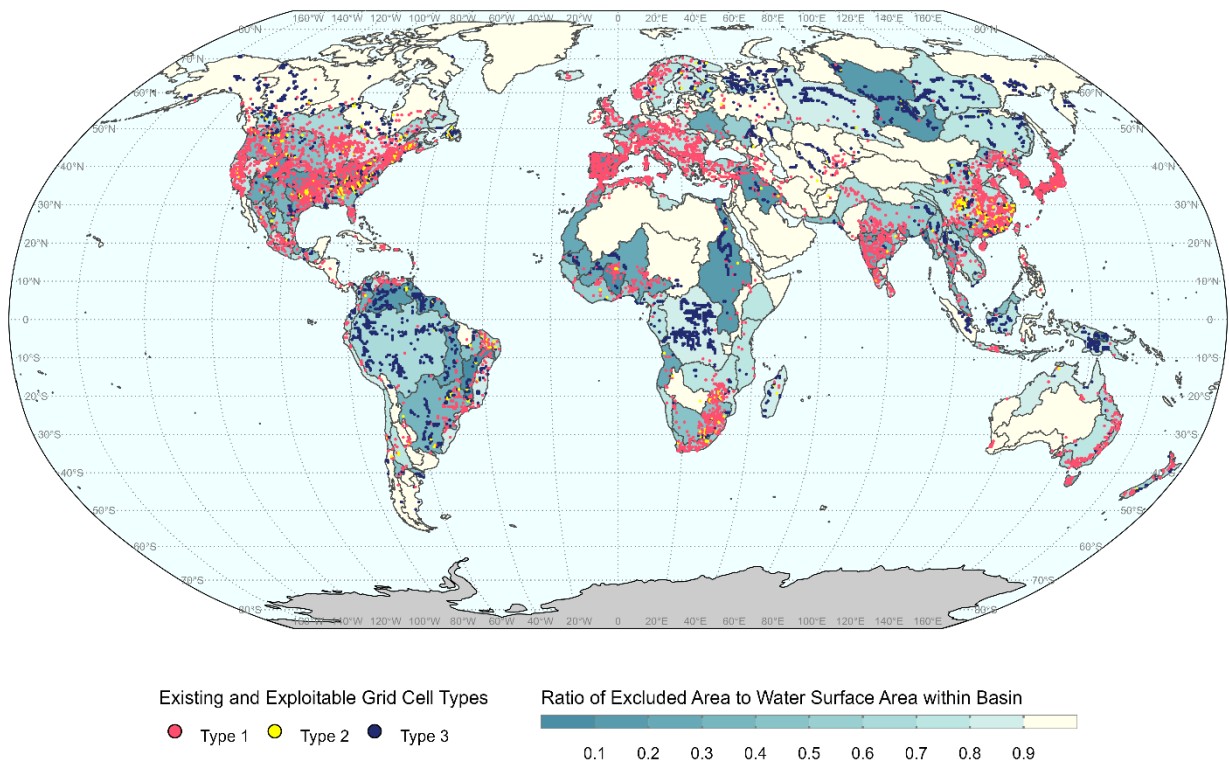


**Figure 5. Global existing and exploitable types (0.5-degree grid cells). Type 1 ("exploited") indicates grid cells that contain existing reservoirs, and in which no additional reservoir expansion is possible. Type 2 ("partially exploited") indicates grid that have existing reservoirs, but in which more reservoirs can potentially be built. Type 3 ("unexploited") indicates grid cells with no existing reservoirs, but in which new reservoirs can potentially be built. All remaining grid cells (not shown on the map) have no reservoir exploitable potential. The color of each basin denotes the ratio of excluded areas to the basin's total water surface area.**


### 2.3.6 Input Layer: Reservoir Construction Cost

We adopt the methodology introduced by Wiberg and Strzepek (2005) to estimate the cost of reservoir storage construction in each basin. The cost of reservoir storage is affected by a diversity of factors such as dam and reservoir size, location, and physiography. Physiographic characteristics are based on various factors, such as slope, topography, vegetation, climate, and soil types. Wollman and Bonem (1971) developed reservoir storage and cost relationships for 11 reservoir size classes in 10 physiographic zones across the U.S. We are not aware of any comparable studies for non-U.S. regions. To enable applications of the storage capacity-cost relationship to other regions, Wiberg and Strzepek (2005) developed a relationship between a physiographic zone and the average slope of the zone. This generalized relationship enables a form of regionalization, where we can use slope as the regressor to assign any non-U.S. region to its nearest corresponding U.S. physiographic zone. In this study, we apply the positive cost-slope relationship for various reservoir size classes (See Table





S2) from Wiberg and Strzepek's study to estimate the normalized unit cost to construct reservoir storage in each of the 235 global basins. The relationships indicate higher slope and smaller storage size will lead to higher unit cost. We re-grid slopes onto 0.5-degree grid resolution using slope data from the Global Multi-resolution Terrain Elevation Data 2010 dataset (GMTED). Basin average slope is calculated as the average of the slopes in the grid cells that fall into grid cell types 1 – 3 (in Fig. 5). The size of the new reservoir is estimated based on existing reservoir sizes within a basin to determine the size class. The normalized unit cost is then multiplied by the average value of unit cost for global reservoirs (Keller et al., 2000) to obtain the scaled units of currency.


To make use of the relationships described in the previous paragraph (i.e., to translate new reservoir capacity in each region into an overnight construction cost), we need to make assumptions about the size of any new increment of reservoir storage capacity. We assume each new increment of reservoir storage capacity, referred to henceforth as the "storage expansion increment", is constant in magnitude for a given basin, therein implying a construction that is identical in size for each new reservoir. In reality, multiple factors affect decisions surrounding the size of new reservoirs, including sub-basin physiography, water sources, economic constraints, non-water supply objectives, hazard concerns (e.g., vulnerability to earthquakes), policy influences, culture, and other factors. As a simplification, we assume dams and reservoirs need to be built on known water bodies including existing reservoirs, rivers, and natural lakes (e.g., based on HydroLAKES). This assumption excludes the sizes of new impoundments that result from expanding the upstream surface area when constructing reservoirs across a river or valley. Thus, the storage expansion increment for a basin is determined as the average size of the existing reservoirs and lakes located in the grid cell types 1 – 3 within the basin. For basins without any exploitable grids, we calculate a replacement value as the mean size of all existing lakes regardless of expansion zone constraints. Finally, depending on the size class into which the storage expansion increment falls, we estimate the normalized unit capital cost (e.g., USD/m$^3$) for storage construction in each basin using equations from Table S2. Figure 6 shows the currency-scaled unit cost of reservoir construction across the global basins. Ultimately, the size of storage expansion increment within a basin plays an important role in determining the curvature of a supply curve, which we will discuss in *Section3.2*.








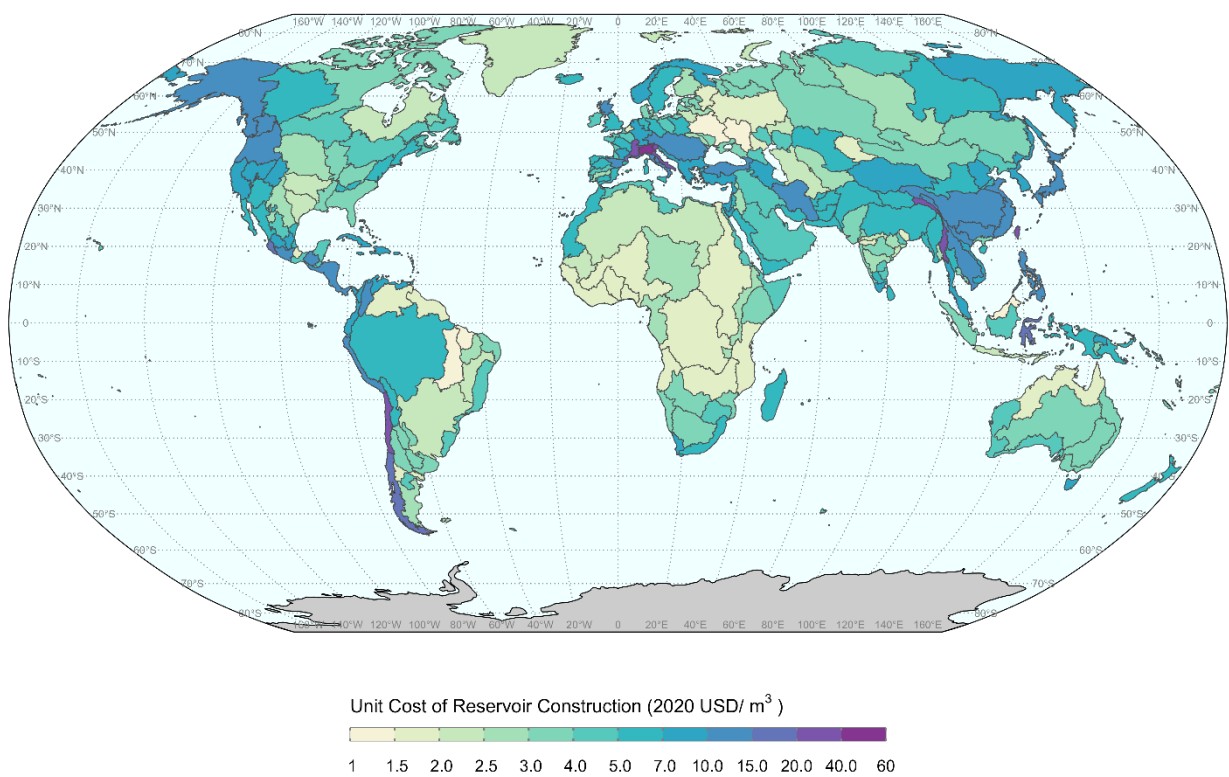

Unit Cost of Reservoir Construction (2020 USD/ m³ )

1    1.5    2.0    2.5    3.0    4.0    5.0    7.0    10.0    15.0    20.0    40.0    60

**Figure 6. Global unit cost estimation of reservoir storage capacity construction in 2020 USD/m³.**

### 2.3.7 Water Supply Curve

A supply curve in GCAM describes the relationship between the quantity of a commodity being supplied (e.g., surface water, or solar power) and the unit cost of that commodity. The renewable water supply curve proposed in this study allows GCAM to grow its reservoir water supply over time through incremental investments in new reservoir storage capacity. Just as electricity supply requires capital investments in new power plants, reliably extracting surface water supply (via reservoir storage) requires investments in reservoir capacity that carry substantial levelized costs. Thus far, we have only described our approach to generating the input elements required to construct a supply curve, rather than detailing the construction of the supply curve itself. Specifically, we described the generation of a capacity-yield curve in *Section 2.3.2*, and the cost to construct reservoir storage in *Section 2.3.6*. In this *Section (2.3.7)*, we describe our approach to combining these two threads of information to construct each basin's renewable water supply curve.

The supply curve covers the entire range of possible supply quantities and associated prices. The base point of every supply curve is at (0.0001, 0), meaning there is a small cost ($0.0001/m³) even if the supply is zero, to account for externalities that occur in the absence of supply. Beyond this point, increasing quantities of supply require increasing levelized costs. This is a common characteristic among renewable resource supply curves (including for electricity technologies such as wind and



solar power). This upward-sloping shape reflects that exploitation of the resource becomes more costly after the best
        components of the renewable flux (e.g., the best regions or portions of the flow-duration curve for extracting flow cheaply)
        are exhausted. The shape a given basin's curve takes depends on the amount of yield gained through reservoir expansion at
        various storage capacity expansion increments versus the corresponding capital costs of that expansion.

        Equations (14) - (17) describe our approach to generating the set of points that define a renewable water supply cost
curve. The objective, captured by Eq. (17), is to identify a set of supply curve points $(Y_j, P_j)$ that define the quantity of water
        supplied $(Y_j)$ and the cumulative levelized cost $(P_j)$ to provide that quantity for cumulative increments of storage capacity $(K_j)$
        expansion. To produce the cost for each new increment of investment in capacity expansion, we calculate the levelized cost of
        storage capacity (LCOSC) (Eq. 16), which is given as the ratio of equivalent annual cost (EAC) (Eq. 14) of each storage
        expansion increment to the yield gained from that increment (Eq. 15). The EAC comes from a capital recovery levelization
that accounts for the cost (described in *Section 2.3.6*) to build reservoir storage in each basin. The gain in yield from storage
        capacity expansion comes from the nonlinear capacity – yield relationships defined in *Section 2.3.2*. A supply curve is then
        constructed by accumulating the yield gain $(\Delta Y_j)$ as well as the corresponding levelized cost $(LCOSC_j)$, shown in Eq. (17),
        until the yield reaches a preliminary end point where no more supply can be reliably provided by reservoir storage. This
        preliminary end point is determined either as the annual runoff or as the maximum water yield from the maximum exploitable
storage capacity ($K_{max}$ is determined in session 2.3.5), whichever is smaller (Eq. 17). If the maximum water yield is smaller
        than the annual runoff, the supply curve can still be extended to reach the annual runoff to indicate the potential for acquiring
        water supply through other (i.e., non-reservoir based) means, such as costly water transfers and transport. To reflect the higher
        cost of these additional renewable water supply measures, we assume the levelized cost for the extended portion on the supply
        curve is five times the reservoir cost.


$$EAC_i = \frac{C_i \times r}{(1-(1+r)^{-n})} + OM \tag{14}$$

$$\Delta Y_i = Y(K_i) - Y(K_{i-1}) \tag{15}$$

$$LCOSC_i = \frac{EAC_i}{\Delta Y_i} \tag{16}$$

$$\begin{cases} Y_j = \sum_{j=1}^{j=i} \Delta Y_j = Y(K_j) \\ P_j = \sum_{j=1}^{j=i} LCOSC_j \end{cases}, Y_j \leq \min\left(I_g, Y(K_{max})\right) \tag{17}$$

where $C_i$ is the 'overnight capital cost' for the expansion storage at the expansion stage $i$; $r$ is the discount rate that determines
the present value for future cash flows, assumed as 0.05; $n$ is the reservoir useful lifetime, assumed as 60 years to be
conservative considering factors like sedimentation; $OM$ is the fixed annual operation and maintenance cost, assumed to be
0.17% of capital cost (Petheram and McMahon, 2019); $Y$ indicates the non-linear relationship between storage capacity and
yield; $K_i$ is the storage capacity at the expansion stage $i$; $\Delta Y_i$ is the yield gain when basin storage capacity increases from $K_{i-1}$
        to $K_i$; $LCOSC_i$ is the levelized cost of storage capacity when increasing from $K_{i-1}$ to $K_i$, defined as the average revenue per





unit water used that would be required to recover the costs of building and operating reservoir facilities during the project lifetime; $P_j$ is the water price at stage $j$; $i$ indicates the $i^{th}$ expansion stage, $i = 1, ..., m$; m indicates the total number of expansion stages as the integer quotient of the "storage capacity exploitable potential" and the "expansion storage"; $j$ indicates

the $j^{th}$ point on the supply curve, $j = 1, ..., i$. We show examples of supply curves as part of our results in *Section 3.2*.

**2.4 Scenario Design**

We designed a set of stylized scenarios (Table 1) to explore two research questions that highlight the types of science questions that can be posed with this new GCAM-GLORY capability: (1) How does the cost to supply water from reservoir storage vary across global river basins, and to what extent is it shaped by hydrologic versus economic characteristics?; and (2)

What insights and dynamics emerge from the new approach, and to what extent can they be attributed to human-earth system feedbacks? Our four scenarios, which successively build on one another, are designed to highlight key differences in water supply-demand dynamics that emerge from our new approach (compared to the existing approach in GCAM). The first two scenarios use GCAM as it exists now, while the latter two scenarios use the new capability presented in this study.

The *Reference* scenario reflects GCAM as it exists now (i.e., without the advances proposed by our study), and thus

establishes a baseline against which scenarios using our new approach can be compared. The *Reference* scenario is a "Business As Usual" (BAU) case, representing a future pathway that extends historical trends for socioeconomics (i.e., consistent with SSP2 population and GDP assumptions), climate (i.e., no climate impacts), technology, and policies (Calvin et al., 2019). The renewable water supply curve in the *Reference* is constructed by implementing the procedure described in *Section 2.2.3*, which includes establishing the accessible water fraction (by applying Eq. 1), using historical hydrologic time series and historical

levels of reservoir storage capacity), as well as the supply curve's maximum value (refers to the average historical annual runoff) in each basin.

The *Climate Impacts* scenario adds climate change impacts (on water availability only) on top of the *Reference* scenario. Specifically, we impose the influence of climate change impacts on monthly regional water balances with global coverage using climate forcing from MIROC-ESM-CHEM (Watanabe et al., 2011) Global Climate Model (GCM) under RCP

6.0 from the ISIMIP 2b (Frieler et al., 2017) simulation round. To build the renewable water supply curve, we establish the accessible water fraction in the same way we did in the *Reference* (i.e., applying Eq. (1), using historical hydrologic time series and historical levels of water storage), but the supply curve's maximum value is different for each time step, calculated using smoothed, climate impacted future runoff time series in each basin. We do not seek here to conduct a nuanced analysis of the implications of climate change, which we leave to a future study. Rather, we developed this *Climate Impacts* scenario because

it enables a more harmonized assessment with our new, GLORY-based approach, which explicitly accounts for climate impacts. Despite the fact that it is perhaps more representative of reality, we did not label the *Climate Impacts* scenario as a *Reference*, because GCAM does not yet account for climate impacts in its default scenario. All of our scenarios, including *Climate Impacts*, extend through 2050, which is approximately when the different RCPs substantially diverge in their





trajectories. Thus, the forcing level appearing in our *Climate Impacts* scenario is reasonably representative of climate forcing
and impacts across different RCPs through 2050.

The latter two scenarios use our new, GLORY-based approach to generating cost curves. Both scenarios are built by ingesting the very same drivers from the *Climate Impacts* scenario, though the climate impacts register differently than in the first two scenarios, because of GLORY's representation of reservoir storage dynamics, capacity expansion, and costs. The *No Feedbacks* scenario develops a new supply curve and passes it to GCAM in each time period, but does not include demand-
based feedbacks from GCAM to GLORY. In contrast, the *Feedbacks* scenario feeds sectoral annual water demand and solved water supply quantity from GCAM to GLORY in each time step (details described in *Section 2.1.1 and 2.3.1*). The difference between GLORY-based scenarios and the *Climate Impact* scenario offers insight into the incremental effect of our new approach to representing reservoir storage in GCAM. Comparison between the *No Feedbacks* and *Feedbacks* scenarios allows us to explore the sensitivity of the methodology (and its resulting GCAM outputs) to the representation of feedbacks. The latter
comparison could offer particularly important insight given the growing interest in multi-model linkages and feedbacks in the multi-sector dynamics literature (Calvin and Bond-Lamberty, 2018; Weyant and Fisher-Vanden, 2020). Implementing such feedback mechanisms takes substantial effort and computing resources, so it is beneficial to understand the effects of including feedbacks in diverse MSD modeling contexts.

**Table 1. Description of scenario design.**

| Scenario | Modeling Approach | Description |
|---|---|---|
| *Reference* | Existing (i.e., original) GCAM supply curve approach | Default GCAM scenario without any changes. Includes historical trends for socioeconomics (i.e., SSP2), climate (i.e., no climate impacts), technology, and policies. |
| *Climate Impacts* | Existing (i.e., original) GCAM supply curve approach | Builds on the *Reference* scenario by including climate impacts on regional water balances (with global coverage) using climate forcing from MIROC-ESM-CHEM GCM under RCP 6.0. |
| *No Feedbacks* | New GCAM supply curve approach (i.e., using GLORY), without GCAM-GLORY feedbacks. | Uses the same climate forcing and variables (e.g., temperature and precipitation) from the *Climate Impacts* scenario, and updates GCAM in each time step with supply curves generated using GLORY (with no GCAM water demand feedbacks) in each time period. |
| *Feedbacks* | New GCAM supply curve approach (i.e., using GLORY), with GCAM-GLORY feedbacks. | Uses the same climate forcing and variables (e.g., temperature and precipitation) from the *Climate Impacts* scenario, updates GCAM in each time step with supply curves generated using GLORY after receiving water demand feedbacks from GCAM in each time period. |





## 3 Results

We begin (in *Section 3.1*) by exploring regional differences in the capacity-yield relationships that emerge from the GLORY model. We show results for the highest fidelity (i.e., closest to reality) representation of the two GLORY-based scenarios from Table 1 (*Feedbacks*). *Section 3.2* evaluates GCAM cost curves, to which capacity-yield relationships are one input (for GLORY-based scenarios). This section explores cost curve differences across regions, time periods, and all scenarios. *Section 3.3* focuses on the regional implications of the new cost curves produced by our highest fidelity scenario (*Feedbacks*), focusing on its impact on the portion of the cost curve that reflects cheaply accessible water. *Section 3.4* focuses on how GCAM water withdrawal results change (across surface and groundwater withdrawals) across scenarios, to highlight the implications of the new approach presented here. Finally, *Section 3.5* uses bivariate analysis and system feedback loop analysis to evaluate which aspects of our new approach (from economics to hydrology) are most strongly shaping GCAM water supply and demand dynamics.

### 3.1 Implications of the Reservoir Capacity – Yield Relationship

The volumetric yield that can be gained from expanding reservoir storage capacity varies across basins. To roughly capture the shape of each basin's capacity-yield relationship, we calculate and plot the median slope of each basin's curve under the *Feedbacks* scenario, to enable comparison (across basins) of the relative effectiveness of building reservoirs to supply water (Fig. 7). The magnitude of the slope is not only driven by the magnitude of annual runoff, but also by the sub-annual characteristics, as well as the magnitude of expanded capacity (i.e., where on the capacity-yield curve a basin is starting). For example, the median slopes in Fig. 7 show that despite the Amazon basin's large-magnitude historical annual runoff (about 5000 km³/year), the Amazon produces less reliable water supply from building each unit of reservoir storage capacity in 2050 than the Indus basin, whose historical annual runoff is an order of magnitude lower (140 km³/year). This is because water demands in the Amazon are much lower than total annual runoff, while the Indus experiences a water deficit for more than half of the year in the absence of reservoirs (see Fig. 4). Thus, in the Amazon, natural streamflow (in the absence of reservoirs) provides more than enough water to reliably meet seasonal demands, without the need for any reservoirs to smooth out natural streamflow variability; whereas in the Indus, each increment of new reservoir storage provides substantial value because demands are large (and temporally out-of-sync) relative to runoff. The amount of reliable supply provided by reservoir regulation is more meaningful when compared to the magnitude of seasonal demand. The capacity-yield relationship reveals the effectiveness of reservoir storage capacity in mitigating socioeconomic drought by storing the surplus during the water-abundant seasons and releasing it during the water shortage seasons.



**Figure 7. A map of 235 global river basins, where the color of each basin reflects the median slope of the capacity – yield curve (i.e., water yield gain per unit increment of reservoir storage capacity) for each basin under the *Feedbacks* scenario in 2050. For each of six select basins with historical socioeconomic drought concerns (see Fig. 3), a chart plots the capacity – yield curve (black line). Each of the six charts also shows the slope of the capacity – yield curve at each storage capacity expansion interval (i.e., green area), representing the changing water yield gain per unit expansion of reservoir storage capacity at different storage capacity expansion stages; and an orange circle showing GCAM calculated water supply and its corresponding storage capacity in 2050 in the *Feedbacks* scenario. The larger median slope value indicates the potential of gaining more water from reservoir storage capacity expansion.**

The effectiveness of expanding reservoir storage capacity in supplying water decreases with continued exploitation of storage capacity. The non-linearity of the capacity-yield curve indicates the changing rate of increased water yield for each unit expansion of storage capacity. The green shapes in the six sub plots (Fig. 7) show examples of the changing rate in yield gain versus capacity expansion (i.e., the slope of capacity-yield curve) for six selected basins with historical socioeconomic





drought challenges (as established in Fig. 4). A slope that is initially steep, but that flattens out at increasing capacity, characterizes the curve for every basin, driven by the optimization formulation (e.g., constraints) described Eq. (2) – Eq. (13). We can roughly divide a capacity-yield curve into three sections based on the magnitude of the slopes on the curve: (1) the

"rapid yield gain" section, where the increase in water availability to the reservoir storage capacity expansion ratio is the highest when a basin starts building reservoirs; (2) the "steady improvement" section, where water yield keeps increasing with storage capacity expansion, but the increasing rate is slowing down; and (3) the "reach the ceiling" section (not shown on the curve), where annual yield reaches a maximum equal to the annual runoff, and stops increasing with increased storage capacity. The specific shape (e.g., slopes of the three segments described above) taken on by the curve for a given basin is driven by

complex interactions between the input datasets (e.g., the shapes of the monthly inflow and demand curves) and the LP constraints.

About 60% of basins depend on reservoir storage capacity to regulate streamflow enough to satisfy the magnitude and timing of demands by 2050 (e.g., the six basins highlighted in Fig. 7). The rest of the basins' demands can be satisfied by the maximum natural yield, which is the volume of annual water that is naturally supplied (via streamflow) by the hydrosphere,

without the regulation of reservoirs. Meeting demand in excess of the maximum natural yield requires expanding reservoir storage capacity to secure additional water supply that is temporally consistent with the timing of demands. Using the water withdrawals calculated by GCAM, we use the capacity-yield curve to back-calculate the corresponding minimum storage capacity that is needed to meet total demand. (Note that the capacity-yield curve represents the optimum water yield from a given capacity. Therefore, the back-calculation using GCAM withdrawal indicates minimum storage capacity required.) The

orange circle in each of the six example basins (Fig. 7) shows the minimum storage capacity needed to provide water for agricultural, domestic, industrial, and other water uses in 2050. (This orange data point reflects the minimum storage capacity required in any given time period; though if existing storage capacity exceeds this minimum, we retain that storage capacity in our analysis). It can be useful to compare across basins the expanded capacity relative to the maximum exploitable potential, and the corresponding water yield relative to the maximum potential water yield, which is the highest yield value on the

capacity-yield curve. For example, the Indus basin is relying on 9.2 km$^3$ of storage capacity to provide about 135 km$^3$ of annual yield in 2050, which is about 96% of the maximum yield according to the hydrology and demand patterns for 2050. On the contrary, the California basin requires only 17% of the maximum yield in 2050 that can be provided from about 1.3 km$^3$ of storage capacity. The Indus basin has much higher water yield gain from every unit of storage capacity expansion than the California basin, and the former can potentially provide more affordable water. As a result, the reservoir capacity potential is

mostly exploited in the Indus basin. The capacity-yield relationship enhances our understanding of the role of reservoirs across basins by translating climate and socioeconomic dynamics into quantifiable water supplies.

## 3.2 Comparison of Water Supply Cost Curves

Water supply cost curves produced by the GLORY model are dynamically updated in each model time period based on evolving hydroclimatic (e.g., reservoir inflow) and socioeconomic (e.g., water demand) conditions. Figure 8 compares the


supply curves from the default GCAM (in black) with those from the GLORY model, both from the *Feedbacks* scenario (in

orange) and the *No Feedbacks* scenario (in green). To enable a normalized comparison of supply curves across different basins

with different runoff magnitudes, Fig. 8 plots the supply volume (x-axis) as a fraction of mean annual runoff. (The default

GCAM supply curve fractions are calculated using the process described in *Section 2.2.3*). We only show the portion of the

supply curves for fractions within 0 to 0.8, where water demands for most basins fall. Supply curves for 2030 and 2050 are

highlighted (with thick lines) in the figure to demonstrate how the shape of the supply curve evolves over time under our new

approach. The default GCAM supply curve is static for each basin throughout the simulation periods for both *Reference* and

*Climate Impacts* scenarios, whereas those from the GLORY model vary by time and scenario.

**Figure 8. Comparison of supply curves under *Reference*/*Climate Impacts* (GCAM default), *No Feedbacks*, and *Feedbacks* scenarios**
**from 2020 to 2050 in selected example basins. The multiple lines with same color under *Feedbacks* and *No Feedbacks* scenarios**





**indicate supply curves for each 5-year period from 2020 to 2050. The thick dashed and solid lines are for 2030 and 2050 and the thin solid lines are for the rest of the periods. We only show the available fraction from 0 – 0.8 on the supply curves to highlight the differences between original and updated curves. The y-axis is limited to a certain range to better visualize the curve variations.**

The variation over time in the shape of the supply curves produced by the GLORY model confirms that shifts in sub-
annual climate and demand (e.g., for the *Feedbacks* scenario) strongly shape annual reservoir supply yield and cost, thus informally validating the value offered by the new methodology we present here. For example, the annual runoff for the Huang He basin only slightly increases 1.7% (Fig. S2) from 2030 (i.e., 93 km$^3$) to 2050 (i.e., 94.6 km$^3$). However, water prices in the Huang He basin increases by almost 60% (from 2030 to 2050) for supplying 70% of the annual runoff in the *Feedbacks* scenario. Higher annual runoff does not guarantee more reliable annual supply because the storage-and-release decisions of
reservoirs are constrained by the monthly inflow, storage capacity, and monthly demands reservoirs are optimized to satisfy. The LP optimization maximizes the total annual water yield subject to monthly demand patterns. In the case of the Huang He basin, demand will decrease from 2030 to 2050 (see Fig. S2), leading to a lower water yield in 2050 from the same size of reservoir capacity (Fig. S6). As a result, the price will be higher in 2050 for the same amount of water supply to reflect market competition, compared to 2030.

Feedback from GCAM to GLORY (in the form of sectoral water demand) in each time period affects the prices of water, especially for basins with high socioeconomic drought intensity. Figure 8 compares supply curves for all simulation time periods between the *Feedbacks* and *No Feedbacks* scenarios. In general, supply curves under the *Feedbacks* scenario have higher prices for the same time period compared to the *No Feedbacks* scenario. Without feedback from GCAM, both the annual water demand magnitude and its disaggregation into monthly fractions are assumed to be fixed at 2020 levels. Given
the demands are held constant, the shifts among supply curves between periods in the *No Feedbacks* scenario are mainly driven by shifts in monthly runoff patterns due to evolving climate conditions, and the response of reservoir storage to those changing conditions. In contrast, under the *Feedbacks* scenario, GLORY directly ingests demand outputs from GCAM, where most supply curves shift up from the *No Feedbacks* scenario. For example, water prices in the California basin almost doubled from the *No Feedbacks* to the *Feedbacks* scenario. With feedback from GCAM in place, monthly demand in the California basin
significantly increased from June to October in all periods when runoff is typically low (Fig. S3), compared to the fixed 2020 demand level (Fig. S4). The complex relationships between net inflow and demand drive the release decisions from reservoirs, where the California basin experiences less water yield in *Feedbacks* scenario (Fig. S6) due to difficulties in meeting increased demand during irrigation seasons. The feedback scheme ensures that the model responds correspondingly to the shifts in supply-demand timing. With the GLORY model, sub-annual climate and socioeconomic dynamics are incorporated in
reservoir management decisions to further inform GCAM.

**3.3 Available Low-Cost Renewable Water**

Prior to evaluating the implications of our new method for GCAM outputs, such as water withdrawals, we discuss the spatial and temporal changes of low-cost renewable water under the *Feedbacks* scenario. This complements the analysis





of Fig. 8 by focusing on just the relatively inexpensive part of the cost curve and its change over time. Low-cost renewable

water is a metric that shows the amount of water that can be provided by reservoirs at an inexpensive price (i.e., assuming $0.001/m^3$), which corresponds to a single point on each basin's supply curve. Affordable water prices can indicate a basin has not invested and expanded reservoirs much, or the basin has well-expanded storage capacity, but the LCOSC is relatively low. Figure 9 shows the fraction of low-cost renewable water over the total runoff in 2050 (Fig. 9a) and the fractional change from 2020 to 2050 across global basins (Fig. 9b) under the *Feedbacks* scenario. We use a fraction, instead of a volume, in Fig. 9 to

normalize the effect of different magnitudes of runoff across basins.

**Figure 9. (a) Fraction of low-cost ($0.001/m^3$) renewable water over total runoff for 2050, and (b) Changes of low-cost fraction from 2020 to 2050 under *Feedback* scenario.**





Most basins experience a reduction in the fraction of low-cost renewable water over total runoff from 2020 to 2050
(Fig. 9b), leading to a lack of affordable water in some basins by 2050. The basins with high reductions are mainly in North America and Eastern Europe, while basins with medium to low reductions are among South America, Africa, Asia, Western Europe, and Australia. The fraction reduction implies the supply curve shifts left, which causes a higher price given the same quantity of water supply. Increased water cost can have differential impacts on the production of each demand sector, depending on the water consumption rate varied across demand types (e.g., crop, livestock, technology), locations (e.g., land
types), and market rules (e.g., trades) in GCAM.

A basin with a lower fraction of low-cost renewable water over annual runoff can be caused by: (1) higher annual runoff; (2) higher unit construction cost; and (3) lower gain in firm water yield from building reservoirs, depending on sub-annual hydrologic and demand quantity and their monthly patterns (see *Section 3.1*). For example, the fraction of low-cost renewable water in the Amazon basin is low (Fig. 9a), while the basin is well-known for abundant water resources. The low
water demand, compared to the inflow, in the Amazon basin does not require reservoirs to heavily regulate streamflow to complement demand that is already met by natural streamflow. Therefore, Amazon's low fraction of low-cost renewable water is mainly driven by low demand quantity and large amount of natural runoff. On the contrary, the California basin has a low fraction of low-cost renewable water for a different reason. The reliable water yield gain for each unit expansion is low (see Fig. 7) because the water deficit during summer (see Fig. 4) is difficult to complement with limited reservoir storage capacity.
Therefore, the California basin experiences high water prices.

## 3.4 Implications of Reservoir Storage Capacity in GCAM

GCAM water withdrawals based on the new supply curves presented in this study better reflect the role of reservoir storage in basins with seasonal socioeconomic droughts. (As defined before, socioeconomic drought occurs when there is water deficit between demand and runoff in the absence of reservoir regulation.) Figure 10 shows water withdrawals by water
sources for six example basins under all four scenarios from GCAM. The *Climate Impacts* scenario does not have significant differences from the *Reference* scenario. Both the *No Feedbacks* and *Feedbacks* scenarios have substantial differences from the *Climate Impacts* and *Reference* scenarios, in terms of total withdrawals and how the supply share is partitioned between surface water and groundwater.





**Figure 10. Water withdrawals by source for *Reference*, *Climate Impacts*, *No Feedbacks* and *Feedbacks* scenarios in six selected basins.**

With our new GLORY-based method in place in the *No Feedbacks* and *Feedbacks* scenarios, the Huang He basin experiences lower total water withdrawals, and specifically lower surface water withdrawals, compared to the *Reference* and *Climate Impacts* scenarios. This occurs because our new approach captures sub-annual discrepancies between natural water





availability and demand, and the cost of storage required to bridge that gap. More groundwater is pumped to supplement

surface water, which will have downstream implications for associated energy usage and emissions. Moving to the Indus basin, water supply will decline by more than half by 2050 in GLORY-based scenarios (relative to the original GCAM approach), where a limited reservoir supply around 100 km³/year requires a large amount of groundwater to meet water demands. However, as water prices increase with groundwater depletion, the affordable groundwater is exhausted around 2040. High water prices in the Indus basin cause considerable decreases in demand. On the contrary, basins like the Africa Northwest

Coast experience more water yield in scenarios with reservoir storage represented by the GLORY model.

Establishing feedbacks between GCAM and GLORY can ultimately impact surface water withdrawals, because feedbacks impact the timing of demands, the level of storage required to achieve a particular level of water supply yield, and thus ultimately the prices to supply water. The California basin uses less surface water and less total water through 2050 in the *Feedbacks* scenario than in the *No Feedbacks* scenario. We also note that the surface water withdrawals do not share the same

trend as either annual runoff or annual demand. For example, the overall annual runoff in the California basin shows a smooth decreasing trend from 2020 to 2050, and the annual demand increases about 10% by 2040 and drops 15% from 2040 to 2050 (see Fig. S2). Rather, the surface water withdrawals go up and down in each time period. This is because the annual runoff and demand are not the only drivers to determine the amount of supply. Instead, their intra-annual patterns play a more important role in determining monthly release so that reservoirs are the most reliable. With feedbacks in place, monthly demand

patterns can be better captured in the GLORY model to optimize the monthly release.

### 3.5 The Compounding Influences of Drivers on GCAM Supply – Demand Dynamics

The previous sections illustrated the direct and indirect impacts of drivers on capacity-yield relationships and water supply curves from GLORY, and water sector dynamics from GCAM, for a few example basins. This section advances our understanding of compounding influences of drivers on the supply and demand dynamics across the global 235 basins using

bivariate analysis and system feedback loop analysis. These analyses are essential for two reasons: (1) they offer diagnostic insights to ensure the model operates as expected; and (2) they identify dominant model processes that merit further enhancement.

Figure 11 shows the paired correlation (above diagonal) and joint density (below diagonal) between each combination of paired variables, and the marginal distribution (diagonal) of each variable for global basins in 2050. We select three main

drivers (i.e., expansion cost, median water gain per unit storage expansion, socioeconomic drought intensity) and two metrics (i.e., water prices and water withdrawals) to explore cause-and-effects relationships. Water withdrawals can be treated as a driver for the GLORY model under the *Feedbacks* scenario. Note that all the variables are normalized using log transformation to reduce the high variance in the data due to different characteristics of basins, and to better compare the changes between scenarios. For pairs involving socioeconomic drought intensity, we excluded cases when values are not applicable to log

transformation, such as zeros, in the linear regression analysis (last column in Fig. 11). The exclusion of zeros also enabled us to focus on the basins using reservoirs as supply sources because socioeconomic drought intensity at 0 means demand can be



met by natural runoff. The basins included in this case have water deficit issues and require reservoirs to regulate water to improve water supply, which yield 70 and 63 basins for the *Feedbacks* and *No Feedbacks* scenarios, respectively.



**Figure 11. Paired density plots (lower half) and Pearson correlation plots (upper half) among drivers and GCAM outputs in 2050 under *Feedbacks* and *No Feedbacks* scenarios for selected basins. Drivers include cost for incremental expansion storage, median water gain per unit storage capacity expansion at GCAM solved supply level, and socioeconomic drought intensity. GCAM outputs include water prices and water withdrawals (or demands). The orange and blue colours denote *Feedback* and *No Feedback* scenario, respectively. The asterisks next to the level of correlation (e.g., three asterisks mean strong correlation). Basins are included for the linear regression for water gain per unit expansion and socioeconomic drought intensity if the water supply is above the maximum natural yield and there is a deficit between sub-annual demand and natural runoff. The maximum natural yield is the amount of natural water that can be provided without any reservoir regulation. All variables are normalized with log transformation and both axes are unitless.**





Our results reveal that multiple drivers combine to influence supply-demand dynamics in combination, rather than demonstrating the presence of a single dominant driver (Fig. 11). Between drivers and metrics, socioeconomic drought intensity and water withdrawals show the highest correlation among all paired variables. The socioeconomic drought intensity increases with increasing water withdrawals. Different from hydrologic drought, socioeconomic drought considers both hydrologic and socioeconomic aspects in identifying drought level regarding water supply. Higher demand means there will be more water deficit occurring for months with low inflow, leading to a direct impact on the storage capacity-yield relationship calculated in the LP model. Water withdrawals and water prices also share strong positive correlations because of the non-linear renewable water cost curves embedded within GCAM. Expansion cost is positively correlated with water prices, confirming the model's skill in demonstrating that constructing reservoirs in basins with less ideal climate and physiographic features could lead to more expensive water supply. In addition, the positive correlation between expansion cost and surface water withdrawals indicates that larger reservoirs (which may be required on larger rivers) may have a higher expansion cost, but can also be more reliable in supplying water.

The feedback from GCAM to GLORY forms a balancing loop revolving around demand dynamics of the integrated water, energy, and land sectors (see Fig. S8). The bivariate analysis shows the positive or negative linkages between drivers and metrics that can transform into a causality chain. A full causal loop is closed with feedback from GCAM to GLORY, where the balancing loop attempts to seek equilibrium status over time. For example, in Fig. S8, although water demand can increase because of the development of the agriculture and energy sectors, this increase in demand can in turn reduce the impact of change (e.g., expanding reservoirs to supply more water) over time due to water affordability. The feedback also increases the sensitivity of water prices to socioeconomic drought intensity (compared to the *No Feedbacks* scenario), as their correlation is stronger in Fig. 11. The feedback would have greater impacts in the long-term view as evolving demand due to activities from other sectors will propagate to decisions on reservoir expansion pathways. For example, Fig. S9 shows alternative pathways of minimum reservoir storage capacity expansion for example basins from the *No Feedbacks* to *Feedbacks* scenario. To meet the dynamically growing demand in the *Feedbacks* scenario, the California basin needs double the storage capacity required in the *No Feedbacks* scenario in 2050. Interestingly, the doubled storage capacity in the *Feedbacks* scenario provides less surface water than the *No Feedbacks* scenario (see Fig. 10). This is because the evolving demand signal in the *Feedbacks* scenario almost doubled compared to demand in the *No Feedbacks* scenario in 2045 (see Fig. S4), leading to higher projected socioeconomic drought intensity that eventually raises water prices. As the California basin expands its reservoir storage capacity, the ability of reservoirs to supply low-cost water is inevitably reduced by the compounding influences of the climate and socioeconomic change.

## 4 Conclusions

We introduce a new multi-model approach to representing the impacts of reservoirs on the cost to supply surface water, and the resulting demand for that water, using the dynamically linked Global Reservoir Yield (GLORY) model and





Global Change Analysis Model (GCAM). GCAM is a multisector model of global climate-land-energy-water dynamics that has recently been extensively used to explore science questions about the future of regional and global water resources, such as the economic impacts of global water scarcity (Dolan et al., 2021), sectoral responses to water scarcity (Cui et al., 2018), future virtual water flows (Graham et al., 2020), and the regional implications of global water scarcity (Giuliani et al., 2022) 895 among others. The representation of reservoir water storage in GCAM has remained limited, and recent large ensemble sensitivity analysis studies have noted the sensitivity of future global water scarcity (and its economic impacts) to GCAM's assumptions about reservoir water storage (Birnbaum et al., 2022; Dolan et al., 2021; Turner et al., 2019).

Here we develop a new model, the GLORY model, that determines (for each of 235 global river basins) the relationship between reservoir water yield and reservoir storage capacity by combining multiple physical and economic 900 dimensions, such as sub-annual variations in hydroclimatic conditions and human water demands. The GLORY model then develops a 'supply curve' that defines the unit cost to supply increasing volumetric quantities of water, by combining the capacity-yield relationship with data on the cost to build reservoir storage and limitations to reservoir expansion (i.e., viable reservoir sites). GLORY passes updated supply curves to GCAM in each model time period and receives water demand information back from GCAM in a delayed two-way feedback loop. We design four scenarios to compare our newly developed 905 water supply curves to GCAM's original supply curves, and their impacts on GCAM outputs. Our results show that our multi-model framework has a marked effect on GCAM dynamics (compared to GCAM as it exists now), including on surface water withdrawals (and, as a result, on groundwater withdrawals) in many basins. This is because our approach captures key sub-annual dynamics, such as the temporal discrepancy between streamflow and water demand in each basin, and regional heterogeneity in the cost to build reservoirs.

Our new method makes three core contributions to the literature. First, it advances the current representation of reservoir water storage in a widely used model (GCAM) by accounting for the various physical and economic dimensions that impact reservoir water supply potential. The strong temporal correlation between precipitation, streamflow, and reservoir storage dynamics (Hou et al., 2022) indicates the importance of considering annual and sub-annual variations of climate conditions in managing water supply. A large annual runoff volume may not guarantee that all demands can be met if the 915 timing of supply and demand do not overlap. Therefore, in this study, we emphasize the importance of exploring the effect of timing and quantity differences between streamflow and demand (i.e., socioeconomic drought) on reservoir water supply. Our results show a strong impact of socioeconomic drought on demands and water prices calculated from GCAM. Considering additional physical dimensions, we also constrain reservoir storage capacity expansion potential by population, protected areas, croplands, and existing surface water networks. With regard to economic dimensions, we account for reservoir construction 920 costs based on the reservoir storage capacity size and physiographic features to differentiate spatial variations of water prices.

Second, to our knowledge, our new approach now represents the only existing global integrated model of climate-land-energy-water dynamics that accounts for future reservoir storage expansion pathways and their potential multi-system impacts. Expansion of reservoir storage capacity in GCAM-GLORY results indicates more investment is required in reservoir infrastructure and maintenance for certain basins, which increases the levelized cost of storage capacity. On the other hand,





through the feedback effect we introduced in this study, the rising water prices will slow down the reservoir capacity expansion for basins with limited expansion potential and low water yield return from expansion. The long-term pathways of reservoir storage capacity expansion that now emerge from GCAM-GLORY analyses can help inform analyses of water resources infrastructure investments, disaggregation of regional storage capacity into gridded storage capacity (e.g., as is done for water demands through Tethys (Khan et al., 2023) and land allocation through Demeter (Vernon et al., 2018)), adaptive water
resources planning, and potential inter-basin transfers under climate and socioeconomic changes.

Third, we advance the state of the science on human-Earth system feedbacks by deploying multi-model feedback linkages, which is a growing area of importance in multi-sector dynamics research. We also perform an initial assessment of the benefits of including feedbacks via *Feedbacks* and *No Feedbacks* scenarios. Our results show that water demand feedbacks have substantial impacts on surface water withdrawals in basins with persistent socioeconomic drought problems. Water
scarcity issues are more sensitive and better captured under the *Feedbacks* scenario, where higher water prices are seen due to inadequate surface water resources or increased demand driven by the energy and land sectors. For instance, the California basin experiences a 20% reduction in surface water withdrawals in the *Feedbacks* scenario, compared to the *No Feedbacks* scenario in 2050. Other than feedback from water demand, future work can include feedback from land use change that informs the estimation of storage capacity expansion potential.

While this study improves the reservoir storage representation in GCAM for global basins, there are future opportunities to enhance GCAM-GLORY, including but not limited to: (1) We currently aggregate distributed reservoirs within a basin into a single virtual reservoir and simplify the reservoir network. This simplification is reasonable in the context of GCAM because of its coarse spatial (basin) and temporal (5-year) resolution. However, the timing and magnitude of inflow and demand vary spatially, and the release of upstream reservoirs to downstream reservoirs can alter the inflow shape.
Integrating GLORY with a global distributed hydrologic model can potentially improve the representation of spatially distributed reservoirs and enable analysis at finer resolution. (2) More feedbacks can be represented using the GLORY framework. For example, land use change projected from GCAM can be used to inform the constraint layers in GLORY to update reservoir exploitable potential. (3) Reservoir unit cost for each basin plays an important role in determining prices on the supply curves. Reservoir unit cost for each basin can be scaled by the local currency of reservoir capital cost (although data
is usually limited) or regional GDP, to improve the regional economic representation, rather than using a global averaged capital cost. (4) GCAM is technically capable of operating at a one-year time step (Zhao et al., 2021). Improving the temporal resolution for the interactions between GLORY and GCAM will enhance our ability to capture extreme events and allow us to create continuous representation of reservoir storage dynamics through time.

In summary, enhancing the explicit representation of feedbacks between reservoir water supply and demand sector in
GCAM is important for better understanding the role of future reservoir storage pathways in shaping the co-evolution of climate-land-energy-water systems. As more regions experience water scarcity issues in the future, reservoirs could play an important role in mitigating drought impacts and improving water security. Representing the role of reservoir storage in the



context of hydrology and economics will enable us to develop a more comprehensive understanding of demand and supply dynamics in a global multi-sector dynamic model.

**Data and Code Availability**

All data and codes for GCAM-GLORY v1.0 are publicly available in the meta-repository on GitHub at https://github.com/JGCRI/zhao-etal_2023_gmd. The meta-repository is also stored with a permanent DOI: https://doi.org/10.5281/zenodo.10211057.

**Author Contribution**

MZ and TBW designed and conceptualized the experiments. MZ and TBW performed analysis. MZ curated data, developed the model code, and performed simulations and validations. MZ, TBW, CRV, and PP set up gcamwrapper and integrated high performance computing resources. MZ, TBW, NTG, SK, MB, AFMKC, and SM contributed to the methodology. MZ and TBW wrote the initial draft. All authors reviewed and edited the manuscript.

**Competing Interests**

The authors declare that they have no conflict of interest.

**Acknowledgements**

This research was supported by the U.S. Department of Energy, Office of Science, as part of research in MultiSector Dynamics, Earth and Environmental System Modeling Program.

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
