# Peer review of "GCAM-GLORY v1.0: Representing Global Reservoir Water Storage in a Multisector Human-Earth System Model"

_Geoscientific Model Development, 2023_

## Author Response (AR1)

**Reviewer 1**

Thank you for the opportunity to revise this manuscript. Our responses that appear below will follow the format described below:

- black = reviewer comments
- blue = our responses to reviewers
- green = new text additions to the manuscript (as part of the revision)

We are sincerely thankful to the reviewers for their insightful comments and constructive suggestions. We have thoroughly addressed each point raised by the reviewers in the form of detailed explanations in this response document and modifications throughout the manuscript and the SI.

**R1.1.** The authors Zhao et al submitted a manuscript to GMD entitled with "GCAM-GLORY v1.0: Representing Global Reservoir Water Storage in a Multisector Human-Earth System Model". They intend to improve the representation of GCAM by developing GLObal Reservoir Yield (GLORY) and coupling to GCAM to enable inclusion of a feedback loop and update of the mechanisms of the reservoir module, specifically the water supply potential of reservoirs for satisfying water use demand. The manuscript is logically structured (esp. in Sect. 2) and well written. The authors clearly describe the rationale of the many decisions that have to be made for developing and running GLORY and coupling it with GCAM. Sometimes this reads by nature a bit lengthy but in fact, this level of detailed description is necessary to understand the structure of the model components. The authors demonstrate the application of the model with 4 scenarios to show the effect of the described model development. Those results are also well described, both by figures (e.g. Fig. 7) and the related text (e.g. around L720). The results show the difference to the original, static reservoir representation. Interpretation and scientific evaluation of the results (e.g. relation to other models) is not focus of the manuscript which is fine for the given manuscript type in GMD that focus on model description. So, I think the authors did a very good job and I have only a few remarks which could make the manuscript even more readable / broaden readership.

Thank you for your supportive comments and suggestions, which we agree will help to enhance the readability of the paper for a broader audience. Also, we agree that the paper is highly detailed, and thus can read a bit long. We have thus tried to eliminate unnecessary detail, but we think that the currently remaining content is required to fully understand the contribution.

**R1.2.** Congratulations to the provided meta-repository on GitHub which I enjoyed exploring.

Thank you for exploring our Github meta-repository. We are committed to supporting open science and we built our meta-repository to share data and codes we developed in this study and provide instructions to guide readers to reproduce our experiment and results.

**R1.3.** L 96 Agree, but for readers of communities that are not so familiar with MSD models, it would be really good to read a broad review about the principles and mechanisms of MSD in general, and also the difference to other approaches like GHMs. The authors relate to a nice tabular overview in the Supplement when it comes to the reservoir representation but I have the impression that such a table alone is not too informative and I would wish to see some additional explanatory text. I think both (overview and differences) could be very valuable to embed your work and broaden up readership. It can certainly be a paragraph in the Supplement, if text space in the main manuscript is the limiting point.

Thank you for your suggestions—we agree. First, we have added clarifying text to the Introduction section to describe what MSD models are, and under that MSD umbrella, what "global MSD models" are. This modified text reads as follows:

The emerging transdisciplinary field of Multi-sector Dynamics (MSD) is well positioned to explore these interactions given its focus on modeling complex systems of systems that deliver services, amenities, and products to society (Reed et al., 2022). Under the MSD umbrella, a sub-class of models simulate the integrated human-Earth system with global coverage by representing the integrated interactions among energy-water-land-climate-socioeconomic systems. While global MSD models are well-positioned in theory to explore multi-system interactions, their representation of reservoirs (especially future reservoir storage expansion) have remained limited (Bell et al., 2014) (See our review of reservoir representation in global MSD models in Table S1). Here we enhance the representation of reservoir storage in a global multi-sector model, the Global Change Analysis Model (GCAM) (Calvin et al., 2019), and demonstrate the scientific insights that can emerge as a result of this addition.

We have also expanded the SI to include commentary on Table S1. We think it is better to keep this text in the SI rather than the main manuscript. This is because we wish to streamline our manuscript (which, as the reviewer noted is already rather long). We also wish to make it clear that the focus of the paper is on advancing GCAM's capabilities, rather than serving as a comprehensive review of all global MSD model capabilities. One reason we do not wish to provide such a comprehensive summary is that there is a large and diverse group of models in this category, and they vary widely in quality and consistency of their water sector documentation, so we do not want to misrepresent the current state of capabilities from those modeling teams. Our new SI text now reads as:

Table S1 provides an overview of the representation of reservoir storage across a representative sample of global multi-sector dynamic (MSD) models designed to explore the interactions among climate, land, energy, water, and socioeconomic (CLEWS) systems from regional to global scales. MSD is an emerging transdisciplinary field that models complex systems of systems that deliver services, amenities, and products to society (Reed et al., 2022). A small subset of MSD models maintain full global coverage (i.e., model the entire world), and contain a diverse set of multi-sectoral CLEWS interactions that differ across models. The GCAM model is the focus of this paper. Table S1 is a representative, rather than exhaustive, list of models intended only to provide a broader context regarding the class of global MSD models that GCAM resides within. While we classify all of the models in Table S1 as global MSD models, a separate but long-standing body of literature also refers to many of these models as "Integrated Assessment Models" (or IAMs; Weyant 2017; Fisher-Vanden and Weyant, 2020). We use the label "global MSD model" here for multiple reasons, including to denote that models such as GCAM have substantially evolved with regard to spatiotemporal and sectoral process resolution, and have placed increasing focus on impacts, adaptation and vulnerability, and have thus evolved substantially enough from the original

simple climate-energy "IAMs" to warrant a new clarifying label (global MSD model). It is also worth noting that each "model" may actually include a whole suite of models designed to interact with one another.

As shown in Table S1, the models share similarities along the "water availability" and "water supply" dimension (see definitions below Table 1). Reservoirs appear most prominently in the water supply category. The models are similar in the sense that they all include (often as part of a broader multi-model framework) a hydrology model (e.g., LPJmL), which in turn may (or may not) represent reservoir storage. While the hydrology models may represent reservoir storage, we find that they often do not represent "reservoir storage expansion", including GCAM. Thus, we believe the current study is a novel contribution to considering global reservoir storage expansion. While not the focus here, the global MSD models differ significantly along the water demand dimension, including whether the process is handled exogenously or endogenously to the core global MSD model, as well as with regard to the approaches (e.g., economic versus physical) for allocating scarce water resources to different demand sectors.

**R1.4.** Sometimes (e.g. the overview section within Sect. 2.1) I read several aspects of review, motivation and application possibilities which I would not necessarily see well fitting in a methods section. Therefore, I would suggest to concentrate to methodological description in the methods section and put other aspects to other parts of the manuscript. Specifically, I was a bit surprised to read in around L 150 two (additional) research questions; would have loved to read those in the Intro section. That could help to streamline the manuscript and avoid repetition. On the other hand, I really enjoy reading Sect. 2.2.1. So, it is a bit a question of the right amount of overview (nicely done in 2.2.1) or going sometimes a bit beyond (e.g. 2.1.1) and of course it is subjective to find the right balance. I nevertheless would suggest to go through and try to improve the balance of the components.

We thank the reviewer for their valuable suggestion. In response, we have revised both the introduction and methodology sections to improve the balance between methodological details and introductory context. Specifically, we have refined section 2.1.1 and enhanced the concluding paragraph of the Introduction section to ensure a comprehensive coverage of the respective sectional topics.

**R1.5.** In some sections, e.g. 2.3.4 it is not always clear which time period is meant. E.g. Fig 4 vaguely expresses "historical SEDI levels" – but it would be good to have a statement in the figure caption which time period is meant. Also, as some of the input data are depending on Xanthos input it would be good to read which climate input the GHM is using (specifically to generate the diagrams in Fig. 4). This certainly would help the reader to digest such diagrams.

Thank you for bringing this to our attention. We have provided clearer explanations regarding the historical period used for calculating the average historical SEDI levels, spanning from 2005 to 2010, both within the manuscript text and in the caption of Figure 4. Additionally, we have enhanced the clarity surrounding the datasets used to drive Xanthos, including the historical climate dataset (WATCH) and the future climate dataset (MIROC-ESM-CHEM under RCP 6.0). The modified caption reads as follows:

Figure 1. Average surplus and deficit between water supply and demand during historical 2005 – 2010 period in six selected basins. The historical demand is from Huang et al. (2018) and the historical inflow is derived from Xanthos runoff driven by WATCH climate forcing (Weedon et al., 2011).

The modified text reads as follows:

Figure 4 shows the average historical (i.e., 2005 - 2010) SEDI levels globally with examples of average monthly water deficit and surplus for six basins.

**R1.6.** L 46 Text in brackets should be within the previous sentence

Thank you. We have relocated the sentence enclosed in brackets to be within the preceding sentence.

**R1.7.** L510 GranD should be read as GRanD

Thank you for spotting this. We have thoroughly reviewed the entire manuscript and corrected instances of the "GranD" to "GRanD."

**R1.8.** L 256 I cannot find the two references (with the indicated year) in the reference list. Please check carefully the coherence of references within the manuscript and the reference list.

We thank the reviewer for identifying this. After a thorough review of the entire manuscript, we have added the references that were cited in the manuscript, but were not included from the reference list:

- Liu, Y., Hejazi, M., Li, H., Zhang, X., and Leng, G.: A hydrological emulator for global applications – HE v1.0.0, Geoscientific Model Development, 11, 1077–1092, https://doi.org/10.5194/gmd-11-1077-2018, 2018b.

- Sanmuganathan, K., Frausto, K., Heuperman, A., Hussain, K., Maletta, H., Prinz, D., Anguita Salas, P., and Thakker, H.: Assessment of irrigation options, Cape Town: Final Draft, WCD Thematic Review Options Assessment IV, 2, 2000.

- Vernon, C. R., Hejazi, M. I., Turner, S. W. D., Liu, Y., Braun, C. J., Li, X., and Link, R. P.: A Global Hydrologic Framework to Accelerate Scientific Discovery, Journal of Open Research Software, 7, 1, https://doi.org/10.5334/jors.245, 2019.

- Zhao, X., Calvin, K. V., Wise, M. A., Patel, P. L., Snyder, A. C., Waldhoff, S. T., Hejazi, M. I., and Edmonds, J. A.: Global agricultural responses to interannual climate and biophysical variability, Environ. Res. Lett., 16, 104037, https://doi.org/10.1088/1748-9326/ac2965, 2021.

**R1.9.** Fig 8: Please write in the figure caption the thick black line.

Thank you for your suggestion. We have revised and included an explanation for the thick black line in the Figure 8 caption:

"The thick dashed and solid lines in orange and blue are for 2030 and 2050 and the thin solid lines are for the rest of the periods. The thick black line represents the supply curve for the *Reference/Climate*

*Impacts* scenarios, which is created from the existing GCAM supply curve approach described in *section 2.2.3.*"

**R1.10.** Fig 9: suggest to extend the figure caption to better grasp the content solely from figure and figure caption.

Thank you for your suggestion. We have enhanced the caption of Figure 9 to better illustrate the concept of the fraction of low-cost renewable water. The new caption reads as:

Figure 9. (a) Fraction of low-cost ($0.001/m^3$) renewable water over total runoff for 2050, and (b) Changes of low-cost fraction from 2020 to 2050 under Feedback scenario. The fraction of low-cost renewable water is affected by runoff amount, reservoir unit construction cost and reservoir yield characteristics. Changes in the fraction of low-cost renewable water represent shifts in the supply curve concerning water price at $0.001/m^3$.

**Reviewer 2**

Thank you for the opportunity to revise this manuscript. Our responses that appear below will follow the format described below:

- black = reviewer comments
- blue = our responses to reviewers
- green = new text additions to the manuscript (as part of the revision)

We are sincerely thankful to the reviewers for their insightful comments and constructive suggestions. We have thoroughly addressed each point raised by the reviewers in the form of detailed explanations in this response document and modifications throughout the manuscript and the SI.

**R2.1.** The authors have developed a reservoir model, GLORY, that can be connected to the global integrated assessment model GCAM. This paper describes GLORY and reports the results of several preliminary experiments when connected to GCAM.

Water use has both physical and economic aspects. To capture water use from an economic aspect, a demand curve and a supply curve are necessary. For water without an explicit market, defining and estimating these is fraught with great difficulty. This study is extremely interesting because it attempts to break through this problem by focusing on the function of reservoirs.

Although the paper is long, it is basically well organized. However, I think there is a lack of description in terms of how the water balance and water use of the vast watershed was modeled. I also think there is a lack of definition of fundamental concepts such as "yield." Consequently, the results, such as producing a "yield" that is more than one order of magnitude larger than the water storage capacity, are not immediately convincing to me, who have studied water use from a physical aspect. I think an additional explanation and information needs to be added.

Thank you for your positive and helpful feedback. We agree that, while our manuscript is highly detailed, two areas in particular would benefit from enhanced explanation: (1) how we reconcile water supplies and usage at basin resolution (for GCAM) while also taking into account finer sub-basin- or reservoir-scale details (GLORY); and (2) how we define reservoir yield specifically, from the level of individual reservoirs to aggregate basin scale. To accommodate this excellent suggestion, we modified the Methods section. Specifically, we have provided detailed explanations regarding these two areas in the response to reviewer comments **R2.3**, **R2.7**, and **R2.21**. We hope that our responses to reviewer and the modifications in the manuscript clarify the design and underlying assumptions of our approach.

**R2.2.** Line 36 "Oki and Kanae, 2006; Vorosmarty et al. 2000; Organization and Fund 2000": These studies pioneered the global water resources assessment at an annual time frame, but little was mentioned on a sub-annual scale. A sub-annual time frame was first introduced by the work of Hanasaki et al. (2008), followed by Wada et al. (2011) and Hoekstra et al. (2012).

We appreciate the reviewer's suggestion to include studies addressing sub-annual freshwater scarcity. We have incorporated these studies into the introduction to provide comprehensive evidence of the water scarcity issue ranging from annual to sub-annual scales across the world. The modified text now reads as follows:

Water exists in relative abundance globally, but its spatiotemporal distribution has historically posed challenges for reliably meeting humanity's water demands. For this reason, over one-third of the world's growing population already faces water shortage for at least one month of each year (Salehi, 2022; Hoekstra et al., 2012; Wada et al., 2011; Hanasaki et al., 2008; Oki and Kanae, 2006; Vörösmarty et al., 2000; Organization and Fund, 2000).

**R2.3.** Line 255: "water users can readily access the mean flow at excess cost. The high cost associated with accessing this upper limit reflects the likely high cost associated with extensive reservoir deployment": This part sounds curious to me. It sounds like water users in the vast basin can access water resources anywhere, anytime. To achieve this, first, one needs to build a reservoir at the outlet (the most downstream point) of a basin to control the runoff. Moreover, a pressured aqueduct network is needed to deliver the outflow to the upstream users (i.e., the reservoir is at the lowest point in the basin; hence, pumping is needed for delivery). I fully understand simplification is unavoidable in modeling, but the treatment within the basin should be clearly mentioned.

Thank you for your comment. The portion of the paper you highlighted illustrates the concept of the *existing* GCAM water supply curve, which differs from our new GLORY-based approach introduced in section 2.3. The existing approach (i.e., without GLORY), described in Kim et al. (2016) and Calvin et al. (2019), assumes that historical annual runoff represents the maximum water available to users in the basin. However, the imposition of a shadow price of $10/m^3$ (i.e., an extremely high price) for the maximum available water effectively means that water withdrawals never reach this upper limit in GCAM (mentioned in Lines 256-257). If water demand within a basin escalates to a level where the price becomes unaffordable, in GCAM the human system adapts by taking measures such as switching to rainfed crops or less water-intensive crops, reallocating crop production to other regions (thus shifting virtual water trade patterns), transitioning to less cooling water-intensive power generation technology, etc. It is important to note that this upper limit of water availability does not presuppose any assumptions regarding reservoir locations, water transfer networks, or reservoir regulations in the existing GCAM water sector. It does effectively assume that water users in each basin cannot re-use water to the point that usage exceeds mean annual runoff. However, as noted earlier, the model encourages various human adaptations before surface water usage and prices reach such high levels.

In contrast, the new GLORY approach developed in this study assumes the integration of all distributed reservoirs into a single unit of virtual reservoir storage capacity. However, we do not assume that the virtual reservoir will be physically located at the outlet. Instead, each virtual reservoir represents an aggregation of individual (distributed) reservoirs that can access all the water resources within the basin. Specifically, each basin's virtual reservoir has a composite subset of characteristics that represent those of the existing distributed reservoirs in the basin. For example, each basin's virtual reservoir has a fractional sub-annual distribution of inflows and evaporation that reflect the specific characteristics of existing (distributed) reservoirs in the basin. While GLORY contains a mass balance for each virtual reservoir that

runs on a monthly "time step", its annual magnitude of reservoir inflow is equal to the annual basin runoff; though, again, the distribution of this annual runoff within each month is adjusted to reflect the sub-annual fractional distribution of inflows to existing reservoir sites.

There are a couple of important caveats to mention regarding this approach. First, each basin's virtual reservoir has an upper limit of annual inflow equal to the average total basin runoff during each five-year window of interest. Establishing a reasonable upper limit is difficult, because in theory all streamflow at any point on a river within a basin could be stored, distributed, returned, and re-used by downstream reservoirs, limited only by the rate of consumptive usage and physical space for reservoirs. Making this assumption would create an enormous upper limit to the water available to be stored and used for irrigation purposes, many times larger than annual runoff. In reality, we rarely observe river basins with rates of irrigation water usage exceeding total basin runoff, both due to high rates of consumption, competition with other sectors for water, and the high prices that would be associated with building storage at less and less favorable sites. (Indeed, as mentioned above, GCAM will initiate human adaptation options in the face of such high water prices). For these reasons, we think that annual runoff represents a reasonable upper limit constraint in our optimization formulation.

A second key limitation of our approach is that we assume each virtual reservoir has characteristics (such as fractional distribution of sub-annual inflow) reflective of existing reservoirs, whereas our broader GLORY-GCAM approach is designed to explore future reservoir expansion. A new fleet of reservoirs would likely create shifts in the composite distribution of reservoir characteristics that are not captured here. Capturing this would require selecting specific new reservoirs and sites, which is beyond the scope of the current study but represents an interesting area for future expansion that is already highlighted in our paper's limitations section.

We have provided further elaboration on these various assumptions and caveats in section 2.3.2 to improve clarity.

GLORY executes 235 unique LP-based optimizations to identify a capacity-yield curve for each of the 235 global river basins in each five-year period, building on the implementation from Liu et al., (2018). It does so by creating a single pool of water storage for each basin that aggregates unifying each basin's distributed reservoirs into a single "virtual reservoir" that reflects the total cumulative basin storage capacity of the basin. Importantly, the spatial assumption underlying the concept of the "virtual reservoir" does not necessarily position the virtual reservoir at the basin's outlet. Rather, we assume that the "virtual reservoir" can access the all the water resources within the basin, enabling collaborative optimization of water supply to meet demand. Additionally, each basin's "virtual reservoir" has a fractional sub-annual distribution of inflows and evaporation that reflects the specific characteristics of existing distributed reservoirs within that basin.

Citations:

Kim, S. H., Hejazi, M., Liu, L., Calvin, K., Clarke, L., Edmonds, J., Kyle, P., Patel, P., Wise, M., and Davies, E.: Balancing global water availability and use at basin scale in an integrated assessment model, Climatic Change, 136, 217–231, https://doi.org/10.1007/s10584-016-1604-6, 2016.

Calvin, K., Patel, P., Clarke, L., Asrar, G., Bond-Lamberty, B., Cui, R. Y., Di Vittorio, A., Dorheim, K., Edmonds, J., Hartin, C., Hejazi, M., Horowitz, R., Iyer, G., Kyle, P., Kim, S., Link, R., McJeon, H., Smith, S. J., Snyder, A., Waldhoff, S., and Wise, M.: GCAM v5.1: representing the linkages between energy, water, land, climate,

and economic systems, Geoscientific Model Development, 12, 677–698, https://doi.org/10.5194/gmd-12-677-2019, 2019.

**R2.4.** Line 263, "The second data point": I guess the authors are discussing the inflection points of the supply curve here. Better to provide a schematic here. Particularly, the second data point is hard to understand.

Thank you for your suggestion. We have indeed included Figure 3 as an example to illustrate the "three key data points used to establish a renewable water supply curve" (mentioned in Line 252). To help clarify that our descriptions of data points are referring to Figure 3, we have inserted clarifications at key points, such as "Thus, the second data point (i.e., accessible point in **Error! Reference source not found.**) on…". This should better guide readers in understanding the process of constructing the supply curve with the visualization in Figure 3.

**R2.5.** Line 269 Equation 1: Better to add a unit of each term. I think the unit of QA is km3 year-1. What I am wondering is the unit for RS. It must be in km3, but then the unit becomes different from the other terms.

Thank you for your suggestion. We have adjusted the description of the equation to include the relevant units ($km^3$/year).

**R2.6.** Line 277 "(Note that… from groundwater depletion observed over a historical calibration period: withdrawal – depletion/runoff)": I am puzzled here. Equation 1 does not include withdrawal. Why do the authors mention water withdrawal here? A further explanation is needed.

Thank you for your comments.

Total basin runoff must be restricted in GCAM to account for the ability of water users to capture, store, and transfer surface water. We refer to this restricted fraction of total basin runoff as the accessible renewable water. The accessible fraction is determined as that which causes GCAM groundwater depletion to match actual, observed groundwater depletion in a historical period. In other words, if we increase accessible renewable water, groundwater depletion goes down, and vice versa; we can therefore calibrate the share of demand met by non-renewable water by adjusting the renewable water accessibility. This analysis assumes negligible desalinated water use currently—a reasonable and practical assumption given the lack of suitable data with global coverage and the fact that the vast majority of basins use negligible desalinated water anyway.

The GCAM data system (Bond-Lamberty et al., 2019) uses two different methods to determine the accessible fraction, depending on the data available for a specific basin. Basins with historical deletion data (about 20% of basins) use *the (withdrawal-depletion)/runoff* equation you referenced in your comment. For basins without historical groundwater depletion data, the GCAM data system will use Equation 1. We have added more details to explain this method.

In basins for which estimates of historical groundwater depletion are available, or approximately one-fifth of basins, the accessible portion of renewable water is estimated with historical annual water withdrawals (GCAM documentation; Hejazi et al., 2014a) and groundwater depletion data (Scanlon et al., 2018), described in Eq. (6). This back-calculation defines accessible water as the maximum annual total water withdrawals from 1990 to 2015 ($QW_i^{max}$) excluding the supply from groundwater depletion ($GD_i$) observed over a historical calibration period. Using these three points (i.e., \$0.00001/m$^3$ for no supply, \$0.001/m$^3$ for the accessible fraction, and \$10/m$^3$ for maximum runoff), *gcamdata* creates a 20-point curve (shown in Fig. 3), with the accessible fraction being the 10th point.

$$QA_i = QW_i^{max} - GD_i \qquad (1)$$

Citations:

Bond-Lamberty, B., Dorheim, K., Cui, R., Horowitz, R., Snyder, A., Calvin, K., Feng, L., Hoesly, R., Horing, J., Kyle, G. P., Link, R., Patel, P., Roney, C., Staniszewski, A., Turner, S., Chen, M., Feijoo, F., Hartin, C., Hejazi, M., Iyer, G., Kim, S., Liu, Y., Lynch, C., McJeon, H., Smith, S., Waldhoff, S., Wise, M., and Clarke, L.: gcamdata: An R Package for Preparation, Synthesis, and Tracking of Input Data for the GCAM Integrated Human-Earth Systems Model, 7, 6, https://doi.org/10.5334/jors.232, 2019.

GCAM documentation: https://jgcri.github.io/gcam-doc/demand_water.html.

Hejazi, M., Edmonds, J., Clarke, L., Kyle, P., Davies, E., Chaturvedi, V., Wise, M., Patel, P., Eom, J., Calvin, K., Moss, R., and Kim, S.: Long-term global water projections using six socioeconomic scenarios in an integrated assessment modeling framework, Technological Forecasting and Social Change, 81, 205–226, https://doi.org/10.1016/j.techfore.2013.05.006, 2014a.

Scanlon, B. R., Zhang, Z., Save, H., Sun, A. Y., Müller Schmied, H., Van Beek, L. P. H., Wiese, D. N., Wada, Y., Long, D., Reedy, R. C., Longuevergne, L., Döll, P., and Bierkens, M. F. P.: Global models underestimate large decadal declining and rising water storage trends relative to GRACE satellite data, Proc. Natl. Acad. Sci. U.S.A., 115, https://doi.org/10.1073/pnas.1704665115, 2018.

**R2.7.** Lie 294 "This yield may far exceed the physical storage capacity of the reservoirs in a basin.": Additional explanation is needed here because the background of this claim is unclear. As shown in Figure 4, the world regions typically have one dry and one wet season in a year. Reservoirs can transfer water from the wet season to the dry season for their storage capacity. First, I am puzzled why the authors claim that the "yield" (same as seasonal water transfer, if I understood correctly) may far exceed the storage capacity. There is another concern: water reuse within a basin. Once water is used, the unconsumed fraction of water is drained, and it can be withdrawn again downstream. Therefore, the volume of water withdrawal can be greater than that of water transferred by reservoirs. If the authors mainly refer to this (i.e., the reuse of water within a basin), it should be clearly mentioned. Note that reuse is constrained by the consumption-to-withdrawal ratio. Hence, I guess it cannot "far exceed" the physical storage capacity.

Thank you for your comments. We define the term "annual yield" here (first mentioned in Line 297) as the annual volumetric quantity of water that can be released from a reservoir (of a given level of storage

capacity) for downstream uses within a year. Our definition conforms to the definition of yield offered in the seminal Loucks and van Beek (2017) text on water resources planning and management. Importantly, our annual yield takes another step beyond the traditional definition, in that it is defined as the sum of monthly releases, and we require those monthly releases to occur in the same pattern as monthly demands (Eq. 6). In a sense, we institute a substantial constraint to maximizing annual yield by requiring that water be distributed exactly in accordance with the pattern of human needs downstream.

In our formulation, as storage capacity increases, so too does yield (hence the capacity-yield curve), with a new curve being defined in each 5-year time period. The water yield from a particular reservoir can surpass its storage capacity because the reservoir regulates streamflow with its storage, and the volumetric quantity of streamflow typically exceeds a reservoir's storage capacity (except for reservoirs that serve in a carryover storage role). In most real reservoirs, the capacity:inflow ratio is less than 1, indicating streamflow exceeds storage capacity. Thus, natural streamflow patterns produce an upper limit on the maximum yield that is possible at any given site, which can only be reached with high levels of storage (Loucks and van Beek, 2017). This is especially notable in basins with high intra-annual streamflow variability, where reservoirs serve to store water during the wet season and release it during the dry season. Natural streamflow also produces a lower limit on yield. By our definition, even with very low (or negligible) levels of reservoir storage capacity, each river basin has the capacity to provide some level of natural annual yield for downstream users in sync with demand patterns. If you refer to Fig. 7, this is why each river basin has a positive yield value on the y-axis even in the absence of reservoir storage. As the paper notes, "the maximum natural yield…is the volume of annual water that is naturally supplied (via streamflow) by the hydrosphere, without the regulation of reservoirs. Meeting demand in excess of the maximum natural yield requires expanding reservoir storage capacity to secure additional water supply that is temporally consistent with the timing of demands."

In summary, central to our definition of yield is the notion that the fundamental value a reservoir provides is to smooth out variability in streamflow that is out of sync with human usage patterns. Thus, we measure the added value of reservoir storage not by its volumetric storage capacity but by the suitability of the reservoir releases (or downstream flows) for human usage.

We have clarified this in the manuscript as follows:

However, the quantity of water that reservoirs effectively supply to meet downstream demand (referred to as the "reliable yield") is not limited to the maximum physical volume of water that can be stored in reservoirs themselves (i.e., their capacity); rather, a key value of storage (in the context of water supply) is the capability to release a reliable yield for downstream use that can consistently meet sub-annually varying demand. The reliable yield is strongly affected by the intra-annual variability of inflow and diverse demands over a year, in addition to the limit from the reservoir's storage capacity. This yield may far exceed the physical storage capacity of the reservoirs in a basin.

Regarding water reuse, in the new GLORY approach, we have considered the water reuse due to the return flow in our assumptions, which is 10% of the monthly release for supply and environmental flow from the "virtual reservoir". This is described in Section 2.3.2.

Citations:

Loucks, D. P. and Van Beek, E.: Water resource systems planning and management: An introduction to methods, models, and applications, Springer, 2017.

**R2.8.** Line 295 "the existing approach": Specify what approach (i.e., studies) the authors refer to. Many physical models can simulate this (e.g., Masaki et al. 2017). Do you mean the earlier versions of GCAM?

Thank you for your comments. We have clarified it as "the existing GCAM's approach".

**R2.9.** Line 341 "by creating a single pool of water storage for each basin": Again, it is better to mention where this aggregated reservoir was located clearly. I guess that the authors assumed that the reservoir is located at the lowermost point of each basin and that redistribution of water within a basin is costless and possible even in the large river basins in the world (e.g., the Nile, the Amazon, the Congo in Figure 1).

Thank you for your comments. We have provided a detailed description of our "virtual reservoir" approach in our response to your question labeled "**R2.3**", pertaining to line 255 in the original manuscript. As a brief summary of our response in **R2.3**, we note that a "virtual reservoir" does not necessarily imply that an aggregated reservoir will be placed at the outlet. Instead, the single virtual reservoir in some ways is a composite reflection of the individual characteristics of distributed reservoirs, though we do use total basin runoff to provide an upper limit to the virtual reservoir's inflow. We have expanded our discussion of this topic in Section 2.3.2. of the manuscript.

**R2.10.** Line 344 "how much yield could be achieved if the system was operated to maximize yield": Same as my comment above. I guess one important assumption of this study is that all runoff generated in a basin flows into the aggregated reservoir, and the outflow from the reservoir can be delivered to the entire basin. Because this assumption is not obvious, it is better to clarify. The concepts of "yield" and "maximization" also require this big assumption.

Thank you for your suggestion. We have provided the detailed explanations in the previous response **R2.3** and **R2.7**.  We have addressed this by clarifying the assumption of "virtual reservoir" in the manuscript:

Importantly, the spatial assumption underlying the concept of the "virtual reservoir" does not necessarily position the virtual reservoir at the basin's outlet. Rather, we assume that the "virtual reservoir" can access the all the water resources within the basin, enabling collaborative optimization of water supply to meet demand. Additionally, each basin's "virtual reservoir" has a fractional sub-annual distribution of inflows and evaporation that reflects the specific characteristics of existing distributed reservoirs within that basin.

We have also clarified the definition of "annual yield" in this study: "the quantity of water that reservoirs effectively supply to meet demand over a year".

**R2.11.** Line 377 Equation 8: I guess $E_t$ is evaporation from the water surface of the reservoir, but the equation looks like the basin's total evapotranspiration. I guess this becomes problematic in arid regions where $I_g \ll E_g$. Under this condition, Equation 3 will become negative. Clarify this point.

Thank you for your comments. As noted in the manuscript, $E_g$ represents the mean annual reservoir evaporation over the GCAM 5-year time step from all distributed reservoirs within a basin, that have a summed storage capacity K, as in $S_{min} \leq S_t \leq K$. $E_g$ is a function of reservoir surface area, which is correlated to the storage capacity (see example in Figure S7). The reviewer is correct that there a few rare cases where $I_g$ is smaller than $E_g$. In such instances, we implemented a maximum allowable multiplier to constrain the reservoir evaporation, preventing it from depleting all the inflow and generating negative values in the yield.

**R2.12.** Line 388: "fp, pt and zt": As a hydrologist, it is hard to imagine that these parameters can be objectively determined. The location, consequently, the catchment area and climate of the reservoir have been changed from reality. Indeed, the authors imply in Line 403 that these parameters were not calibrated against observations. Further elaboration is required on how the authors determined these parameters for 235 basins in the world at a monthly interval.

Thank you for your comments. The parameters $f_t, p_t, and\ z_t$ represent average monthly profiles of annual water demand, inflow, and evaporation at the basin scale. Indeed, the real monthly profiles would vary across locations and with changing climate. As we mentioned above, the "virtual reservoir" is assumed as a collection of distributed reservoirs within the basin, rather than a single virtual reservoir that is placed at the outlet. Our monthly profile estimation is intended to reflect the sub-annual characteristics of inflows to and evaporation from these distributed reservoirs, while complementing the coarser resolution for GCAM in two respects. To clarify this, we added the following text in the manuscript:

Monthly profile of inflows to, and surface potential evaporation from, the virtual reservoir is derived from the Xanthos model's monthly streamflow time series output at grid cells with existing reservoirs for the five-year GCAM period of interest. We consider this approach as a middle ground to address fine resolution from hydrologic variables and coarse resolution from GCAM in terms of two key aspects. First, the monthly profiles evolve with the inter-annual and intra-annual variability of the changing climate. Second, we initialized the inflow and evaporation profiles, focusing specifically on the grid cells with reservoirs to capture hydrologic patterns in reservoir-located areas. The monthly profiles for a virtual reservoir are calculated as the averaged profiles from these gridded inflows and evaporation.

A more advanced but computationally intensive approach, which we leave for a future study, would be to site new individual storage reservoirs over time in individual grid cells. The purpose of the current study is to explore the merits of a lumped virtual reservoir approach. As mentioned earlier, we do propose this more detailed approach as one option for future expansion of our initial study.

**R2.13.** Line 407: "possible for that same level of storage capacity": Same to what? What does "possible" mean?

Thank you for raising this question. When we mention the "same level of storage capacity," we are referring to keeping the storage capacity from GLORY model consistent with historical reservoir storage capacity level aggregated within each basin when comparing the releases. Then, GLORY model calculates the maximum annual water supply (i.e., annual yield) with the historical climate data, estimating the maximum amount of water that can be reliably provided from this level of reservoir storage capacity.

This makes a fair comparison with the historical water demand data that is observed based on the existing historical reservoirs. We have modified the sentence to remove "possible" and clarified that the storage capacity is the historical existing storage capacity. The modified text reads as:

The first aspect is to check that the level of annual water supply achieved with existing reservoir storage capacity (i.e., historical water demand data) is less than or equal to the maximum annual release (i.e., yield) that GLORY suggests for that same volume of the existing historical storage capacity for a given basin. We confirm this finding in Fig. S10a with global water demand data (Huang et al., 2018), demonstrating that our approach provides a reasonable upper bound on the water supply, without attempting to represent each basin's unique water management behavior.

**R2.14.** Line 410: "to check the level of reservoir annual release from existing reservoir storage capacity is within the range of the annual release (same as yield) that GLORY produces at the same level of storage capacity.": This part is particularly hard to understand. Fist, what does "the level" indicate? Second, I believe storage capacity and release are independent. For instance, the storage capacity of the Big Bend Dam (2.2 km3) and the Oahe Dam (28.5 km3) in the Missouri River is different, but their annual release is almost the same because these two dams are cascading. Simply, the mean annual release of a reservoir is the same as its inflow. A further explanation is needed here. Third, again, what does the "same level of storage capacity" mean? What does "level" indicate? Same to what?

Thank you for your comments. We intended to use "the level" to represent that in the GLORY model simulation, the input virtual reservoir storage capacity K (Eq. 6) should be the same as the storage capacity that the reservoir release data is based on for a given basin. The water demand and reservoir release are the two aspects we use to validate the water yield simulated by the model. The annual water yield is the sum of the monthly releases over a year within the GCAM model. Thus, we compared water yield with reservoir release data (from a separate dataset) and confirmed that the model has an acceptable performance. In addition, water demand should be less than the total annual water release, because the total release is an upper boundary for demand. We also validated the model from this perspective with historical water demand data. We have improved the text for better clarity:

The second aspect is to check that the amount of reservoir annual release from existing reservoir storage capacity (i.e., historical reservoir outflows) is within the range of the annual release (same as yield) that GLORY produces at the same volume of the existing storage capacity.

**R2.15.** Line 433 "A basin's optimization for a particular GCAM period (e.g. 2050) is executed in the absence of any carryover of information about reservoir storage levels…": Then, how was the initial storage of reservoir determined at the beginning of a given GCAM period?

Thank you for your question. We use Eq. 5 to assume a steady state condition within a 5-year period. The linear programming model iterates to find solutions until the steady state solution is obtained. Thus, there is no need to assign an initial reservoir storage.

**R2.16.** Line 453 "via a monthly water demand fraction profile that shifts close in appearance to the monthly irrigation demand profile": Hard to read. Do you mean that the profile of the total water demand becomes closer to the new irrigation demand? The expression "shifts close in appearance" sounds highly subjective and arbitrary.

Thank you for your comments.

Basically, we mean to say that if GCAM projects a larger (increasing, relative to other sectors) share of irrigation water demands over time, GLORY will perform a weighted shifting of the distribution of monthly water demands so that they more closely resemble those of the irrigation sector (to reflect the larger role the sector plays in total demands).

To provide additional details, GCAM only provides annual sectoral water demand, rather than monthly sectoral demand. When GCAM-GLORY is in the *Feedback* mode within a future time step, GLORY will first calculate the monthly sectoral demand based on the historical monthly sectoral demand profile and the annual sectoral water demand calculated from GCAM. Then, GLORY will aggregate six sectoral demands by month to get the monthly demand profile for the given time step. This ensures the change in annual sectoral water demand (or the sector demand weight) in each time step can propagate to the monthly demand profile. We modified the text to improve the clarity:

For example, if GCAM projects that future annual irrigation water withdrawals will disproportionately increase over time in a particular GCAM scenario and time period, this information will translate into corresponding disproportionate increase in monthly irrigation water demand compared to other sectors. Consequently, the total monthly demand will be predominately influenced by irrigation water demand. As a result, the profile of monthly water demand fraction will closely resemble that of monthly irrigation demand profile.

**R2.17.** Line 490 "we also filter out grid cells that do not contain existing water bodies suitable for siting of reservoirs (i.e., rivers)": What do you mean "water bodies"? What kind of information the grid cells include? What data did you use?

Thank you for your comments. We defined water bodies and referred to the dataset in the later text within the same paragraph (Line 521-522): "Water bodies are identified using Global Lakes and Wetlands Database (GLWD) Level 3 (Lehner and Döll, 2004), including land types 1, 2, and 3 (lakes, reservoirs, and rivers, respectively).".

**R2.18.** Line 600 Equations 14-17: It is better to show the units for variables.

Thank you for your suggestion. We have included the units for the variables.

**R2.19.** Line 633 "a nuanced analysis": What does this mean?

Thank you for your comments. We have changed the sentence to:

We do not seek here to conduct a detailed analysis of the implications of climate change, which we leave to a future study.

**R2.20.** Figure 7: I expected the unit of Yield is km3 year-1. As mentioned above, I still could not understand why the yield could exceed the storage capacity. To begin with, add a concrete definition of reservoir yield at the beginning of this paper.

Thank you for your comments. The reviewer is correct in noting that the unit of water yield is in $km^3$/year. We have included this unit in the caption of Figure 7. Regarding the question on the reservoir water yield, please refer to our responses elsewhere, particularly in comment **R2.7** and **R2.21** below.

**R2.21.** Line 721: "For example, the Indus basin is relying on 9.2km3 of storage capacity to provide about 135km3 of annual yield in 2050": Again and again, I still cannot understand this. How one can serve 135 liters of water by 9.2 liters of a tank? Also, I am totally lost in how the authors treat the reuse (cascading use) of water within a basin. The authors' model seems theoretically sound from the economic perspective. I am wondering whether the parameters and output numbers can be supported from the hydrological (physical) perspective.

Thank you for your comments. As we mentioned in our response to comment **R2.7**, the storage capacity is the maximum volume of water that a reservoir can store under normal operating conditions. Most reservoirs globally have a capacity:inflow ratio less than one, meaning that total volume of inflow entering a given reservoir on average during a time period of interest exceeds the reservoir's volumetric capacity to store water. The larger the capacity:inflow ratio, the more flexibility a reservoir has to carry over water across seasons or even across years and release it for downstream use. Our previously provided definition of yield, which is consistent with that provided in Loucks and van Beek (2017), is that it represents the amount of water that can be expected to be *released* from a reservoir to meet human demands. The amount of water released from a reservoir is a function of both inflows and the storage in the reservoir itself. By our definition, even with very low (or negligible) levels of reservoir storage, each river basin has the capacity to provide *some* level of annual yield for downstream users in sync with demand patterns. If you refer to Fig. 7, this is why each river basin has a positive yield value on the y-axis even in the absence of reservoir storage. As the paper notes, "the maximum natural yield…is the volume of annual water that is naturally supplied (via streamflow) by the hydrosphere, without the regulation of reservoirs. Meeting demand in excess of the maximum natural yield requires expanding reservoir storage capacity to secure additional water supply that is temporally consistent with the timing of demands."

**R2.22.** Line 746 "informally validating the value offered by the new methodology we present here": What does "informally" mean? How one can "validate" values?

Thank you for your comments. Here we meant only to say that we think the results confirm the value of our methods. In other words, that the sub-annual scale of supply (i.e., runoff) and demand, which our method is built to handle, appears to be quite important. (We did not want to call this a "validation", since

this word has specific meaning in hydrology, so we called it "informal validation". Anyway, we have modified the sentence as follows to improve the clarity:

The variation over time in the shape of the supply curves produced by the GLORY model confirms that shifts in sub-annual climate and demand (e.g., for the *Feedbacks* scenario) strongly shape annual reservoir supply yield and cost. This confirms the added value of our specific focus on discrepancies between sub-annual supply and demand relative to previous studies of global reservoir supply potential (e.g., Liu et al., 2018a).

**R2.23.** Line 766 "The complex relationships between.. due to difficulties in meeting increased demand during irrigation seasons": I guess what the authors report here is that the GCAM-GLORY can tell that irrigation in California should be decreased because irrigation water is unavailable (too costly) for a limited number of months. This is a significant advancement in global hydro-economic modeling.

We are thankful for this positive feedback. We do think that that our feedbacks scenario represents a significant advance, because we are informing future reservoir expansion and operation patterns with water demand data emerging from an internally consistent model that integrates climate-land-energy-water-socioeconomics.

**R2.24.** Line 776 "a single point on each basin's supply curve": What is "a single point"?

Thank you for your question. This single point we refer to is the "accessible point" on a supply curve, when the corresponding water price equals $0.001/m^3$. See Figure 3 and Section 2.2.3. We have clarified this issue in the main text as follows:

Low-cost renewable water is a metric that shows the amount of water that can be provided by reservoirs at an inexpensive price (i.e., assuming $0.001/m^3$), which corresponds to the "accessible point" (explained in *Section 2.2.3*) on each basin's supply curve.

**R2.25.** Line 776 "Affordable water prices can indicate a basin has not invested and expanded reservoirs much, or the basin has well-expanded storage capacity, but the LOSC is relatively low.": Hard to understand. What do you mean here?

Thank you for your comments. There are two different scenarios where water provided by reservoir is more affordable, based on how the supply curve is constructed with Eqs. 15 – 18.

$$EAC_i = \frac{C_i \times r}{(1-(1+r)^{-n})} + OM \tag{2}$$

$$\Delta Y_i = Y(K_i) - Y(K_{i-1}) \tag{3}$$

$$LCOSC_i = \frac{EAC_i}{\Delta Y_i} \tag{4}$$

$$\begin{cases} Y_j = \sum_{j=1}^{j=i} \Delta Y_j = Y(K_j) \\ P_j = \sum_{j=1}^{j=i} LCOSC_j \end{cases}, Y_j \leq \min\left(I_g, Y(K_{max})\right) \tag{5}$$

In the first scenario, if a basin initially has no reservoirs, or just beginning to construct new reservoirs, this implies that there is not a significant accumulation of investments affecting the water price. This concept is illustrated in Eq. 18, where $P_j$ could be small due to limited number of reservoirs being built ($i$ is small).

In the second scenario, the sub-annual variability of inflow in a basin may align well with demand timing. If a reservoir is constructed, the potential annual yield gain could be substantial. This results in high $\Delta Y_i$, leading to a low $LCOSC_i$, according to Eq. 17. Thus, even with additional reservoir expansion within this basin, the sum of LCOSC up to the i-th reservoir ($P_j = \sum_{j=1}^{j=i} LCOSC_j$) could remain small.

**R2.26.** Line 792 "firm water yield": what's this?

Thank you for your question. The phrase "firm yield" (often interchangeably referred to as "safe yield") is the maximum flow that can be made available via reservoir storage given its regulation of the historical streamflows (Loucks and van Beek, 2017). Each increasingly level of reservoir storage capacity has a different (higher) reliable yield. Just how "safe" a given storage capacity's yield is depends on the length of the historical record and the nature of the hydrologic variability captured in it. In general, the "reliability" of a yield is used to indicate the percentage of the time it is expected to be true. Given we are not doing such probabilistic analysis here, we have removed the word "reliable" from our mention of yield, and have also removed "firm yield" from our manuscript.

In our view, it would be quite interesting to expand our analysis in the future to explore the implications of differing yield reliability levels, in effect producing new capacity-yield and supply curves reflecting differing levels of reliability. We have added the following note in the manuscript's limitations section:

(5) Our study explores a very limited sampling of hydrologic uncertainty (i.e., that which is embedded in the historical record and in a single future projection of climate impacts on runoff). It would be of value to develop capacity-yield and supply curve relationships in GLORY corresponding to differing levels of reliable yield (Loucks and van Beek, 2017), as increasingly reliably yields (e.g., moving from 90% to 99%) will require much higher levels of storage investment (and thus costs) to meet demands at those corresponding levels of reliability.

**R2.27.** Figure 10: Surface water withdrawal in the Indus River exceeds 300 km3 year-1 for the Reference and Climate Impact scenarios, which is greater than the mean annual runoff (140 km3 year-1). How could this happen? For the No Feedback and Feedback scenarios, the usage of groundwater will sharply decrease after 2030. How could this be achieved? All in all, what will be the global total water withdrawal in 2050 for each scenario? What will be the global total reservoir capacity for each scenario?

Thank you for your feedback. The *Reference* and *Climate Impacts* scenarios use the *existing* GCAM supply curve approach (i.e., the "old" approach), wherein for Indus River basin, historical groundwater depletion and historical demand is used to calculate the accessible renewable water based on Eq. 2:

$$QA_i = QW_i^{max} - GD_i \qquad (6)$$

Calculating accessible water with Eq. 2 sometimes results in an improper amount of accessible water that exceeds the mean annual runoff. We have identified this inconsistency issue between data and GCAM

results in the current GCAM model, which needs improvement. With our new GLORY approach applied in *Feedbacks* and *No Feedbacks* scenarios, we have addressed this issue. Consequently, surface water withdrawals will always remain below the mean annual runoff.

Regarding non-renewable groundwater, GCAM assumes that non-renewable (i.e., fossil) groundwater will deplete irreversibly. Groundwater availability is categorized into several grades (e.g., grade 1, grade 2, …, grade 10), with each grade permitting a certain amount of groundwater extraction at a specific price. As groundwater is depleted, it moves to higher grades where the price increases. Eventually, groundwater may become fully depleted or the price may become unaffordable. This is documented in Turner et al. (2019). From Figure 10, there is a significant decline in groundwater extraction after 2025 for the Indus basin in the *Feedbacks* and *No Feedbacks* scenarios due to substantial extraction between 2020 and 2025. This depletion results in groundwater moving to higher-priced grades for future extraction. Ultimately, the Indus basin is one in which we observe the potential for a near-term peak and eventual decline in groundwater extraction in a recent study by our team (Niazi et al., in press), owing to groundwater's decreasing levels and thus economic accessibility in the Indus basin.

We have created Figure 1, illustrating the global total water demand from 2020 to 2050 across four scenarios. The future global total water demand shows a decrease of over 1000 km$^3$/year from the previous approach (i.e., used in *Reference* and *Climate Impact* scenarios) to the new GCAM-GLORY approach (i.e., used in *No Feedback* and *Feedback* scenarios). Given that the socioeconomic assumptions remain consistent across all four scenarios in GCAM, the global total water demand remains relatively stable, with notable changes in the proportion of water supply sources (groundwater vs. surface water) in the new approach. In summary, the projected global total demand is approximately 5254, 5185, 4373, and 4356 km$^3$/year by 2050 for the *Reference, Climate Impact, No Feedback*, and *Feedback* scenarios, respectively. The water supply reservoir storage capacity has increased by about 70 km$^3$ by 2050 from historical existing storage capacity under *No Feedback* and *Feedback* scenarios, while it was assumed to remain constant for both *Reference* and *Climate Impact* scenarios. Please note that the increased capacity only pertains to water supply reservoirs, and the projected capacity expansion can vary significantly under different socio-economic drivers (e.g., population and GDP) in GCAM.

[Figure]

Figure 1. global total water demand from 2020 to 2050 across four scenarios.

References:

Turner, S. W. D., Hejazi, M., Yonkofski, C., Kim, S. H., and Kyle, P.: Influence of Groundwater Extraction Costs and Resource Depletion Limits on Simulated Global Nonrenewable Water Withdrawals Over the Twenty-First Century, Earth's Future, 7, 123–135, https://doi.org/10.1029/2018EF001105, 2019.

Niazi, H., Wild, T.B., Hejazi, M., Graham, N.T., Turner, S.W.D, Lamontagne, J., Zhao, M., Msangi, S., Kim, S. (in press). Groundwater Depletion Rates Peak Globally over the 21st Century. Nature Sustainability. DOI: https://doi.org/10.1038/s41893-024-01306-w.

**R2.28.** References

Hanasaki, N., Kanae, S., Oki, T., Masuda, K., Motoya, K., Shirakawa, N., Shen, Y., and Tanaka, K.: An integrated model for the assessment of global water resources - Part 2: Applications and assessments, Hydrol. Earth Syst. Sci., 12, 1027-1037, 10.5194/hess-12-1027-2008, 2008.

Hoekstra, A. Y., Mekonnen, M. M., Chapagain, A. K., Mathews, R. E., and Richter, B. D.: Global Monthly Water Scarcity: Blue Water Footprints versus Blue Water Availability, PLoS One, 7, e32688, 10.1371/journal.pone.0032688, 2012.

Masaki, Y., Hanasaki, N., Biemans, H., Müller Schmied, H., Tang, Q., Wada, Y., Gosling, S. N., Takahashi, K., and Hijioka, Y.: Intercomparison of global river discharge simulations focusing on dam operation— multiple models analysis in two case-study river basins, Missouri–Mississippi and Green–Colorado, Environmental Research Letters, 12, 055002, 10.1088/1748-9326/aa57a8, 2017.

Wada, Y., van Beek, L. P. H., Viviroli, D., Dürr, H. H., Weingartner, R., and Bierkens, M. F. P.: Global monthly water stress: 2. Water demand and severity of water stress, Water Resources Research, 47, W07518, 10.1029/2010wr009792, 2011.

Thank you for these valuable references.